# Downregulation of macrophage Irs2 by hyperinsulinemia impairs IL-4-indeuced M2a-subtype macrophage activation in obesity

Tetsuya Kubota[1,2,3,4,5], Mariko Inoue[1,3], Naoto Kubota[1,2,3,6], Iseki Takamoto[1], Tomoka Mineyama[1], Kaito Iwayama[7], Kumpei Tokuyama[7], Masao Moroi[4], Kohjiro Ueki[1], Toshimasa Yamauchi [1] & Takashi Kadowaki [1,8,9]

M2a-subtype macrophage activation is known to be impaired in obesity, although the underlying mechanisms remain poorly understood. Herein, we demonstrate that, the IL-4/Irs2/Akt pathway is selectively impaired, along with decreased macrophage *Irs2* expression, although IL-4/STAT6 pathway is maintained. Indeed, myeloid cell-specific *Irs2*-deficient mice show impairment of IL-4-induced M2a-subtype macrophage activation, as a result of stabilization of the FoxO1/HDAC3/NCoR1 corepressor complex, resulting in insulin resistance under the HF diet condition. Moreover, the reduction of macrophage *Irs2* expression is mediated by hyperinsulinemia via the insulin receptor (IR). In myeloid cell-specific *IR*-deficient mice, the IL-4/Irs2 pathway is preserved in the macrophages, which results in a reduced degree of insulin resistance, because of the lack of IR-mediated downregulation of *Irs2*. We conclude that downregulation of *Irs2* in macrophages caused by hyperinsulinemia is responsible for systemic insulin resistance via impairment of M2a-subtype macrophage activation in obesity.

[1] Department of Diabetes and Metabolic Diseases, Graduate School of Medicine, The University of Tokyo, Tokyo 113-8655, Japan. [2] Laboratory for Intestinal Ecosystem, RIKEN Center for Integrative Medical Sciences (IMS), Kanagawa 230-0045, Japan. [3] Department of Clinical Nutrition, National Institute of Health and Nutrition, Tokyo 162-8636, Japan. [4] Division of Cardiovascular Medicine, Toho University Ohashi Medical Center, Tokyo 153-8515, Japan. [5] Analysis tool development group, Intestinal microbiota project, Kanagawa Institute of Industrial Science and Technology, Kanagawa 213-0012, Japan. [6] Department of Clinical Nutrition Therapy, University of Tokyo, Tokyo 113-8655, Japan. [7] Graduate School of Comprehensive Human Sciences, University of Tsukuba, Tsukuba 305-8577, Japan. [8] Department of Prevention of Diabetes and Lifestyle-Related Diseases, Graduate School of Medicine, The University of Tokyo, Tokyo, Japan. [9] Department of Metabolism and Nutrition, Mizonokuchi Hospital, Faculty of Medicine, Teikyo University, Tokyo, Japan. These authors contributed equally: Tetsuya Kubota, Mariko Inoue, Naoto Kubota. Correspondence and requests for materials should be addressed to N.K. (email: nkubota-tky@umin.ac.jp) or to T.K. (email: kadowaki-3im@h.u-tokyo.ac.jp)

Obesity is known to cause chronic low-grade inflammation, resulting in insulin resistance and type 2 diabetes[1,2]. Chronic low-grade inflammation is recognized as being caused by adipose tissue macrophage (ATM) accumulation, which has been shown to increase with body weight gain in both humans and rodents[3–8]. Thus, macrophages (MΦs) *per se* are, at least in part, responsible for the insulin resistance observed in obesity. MΦs can be divided into two major populations; the M1-type MΦs, representing the classically activated MΦs, which are activated by Th1 cytokines to generate proinflammatory cytokines, and the M2-type MΦs, representing the alternatively activated MΦs, which are activated by Th2 cytokines to generate anti-inflammatory cytokines[9]. ATMs from lean mice showed high expression levels of the M2-type MΦ marker genes. These expressions were found to be decreased under the high-fat (HF) diet condition[10], associated with aggravation of the insulin resistance[11,12]. M2-type MΦs are classified at least in three sub-populations, M2a-subtype, M2b-subtype and M2c-subtype, based upon the inducing agent and molecular marker expression. M2a-subtype MΦs activated by IL-4 or IL-13 promote of Th2-type inflammation against enhanced fibrosis and wound healing by increased *arginase1*(*Arg1*), FIZZ1 (also called *Retnla*) and *Ym1* (also called *Chi3l3*) expression levels. M2b-subtype MΦs are induced by combined exposure to immune complexes with Toll-like receptor (TLR)- or IL-1 receptor (IL1R)-ligands, whereas M2c-subtype MΦs are induced by IL-10. M2b-subtype MΦs with concomitant high IL-10 and low IL-12 suppress inflammatory cytokines and molecules involving in lymphocytic activation. M2c-subtype MΦs are thought to be predominantly responsible for negative/deactivating immunoregulation[13,14].

In the adipose tissue, eosinophils are recognized as a source of IL-4 production, and eosinophil-deficient mice showed insulin resistance and impaired glucose tolerance under the HF diet condition, with a decrease of IL-4 production and number of M2a-subtype MΦs[15]. Decreased M2a-subtype MΦ activation by IL-4 renders mice susceptible to diet-induced obesity and glucose intolerance[16]. These data suggest that M2a-subtype MΦ activation by IL-4 is impaired in obesity, resulting in obesity-induced insulin resistance.

IL-4 binds to two types of receptors; the type I receptor, which is composed of IL-4 receptor (IL-4R)α and the common γ chain (γC), and the type II receptor, which is composed of IL-4Rα and IL-13 receptor (IL-13R)α1[17]. IL-4Rα binds janus kinases (JAK)1, which is crucial for the phosphorylation of signal transducer and activator of transcription 6 (STAT6). IL-4-induced STAT6 phosphorylation was necessary for M2a-subtype MΦ activation. In fact, M2a-subtype MΦ activation by IL-4 was impaired in systemic *STAT6*-deficient mice and myeloid cell-specific *IL-4Rα*-deficient mice[16,18,19]. In addition to STAT6, Irs2, which was previously called IL-4-induced phosphotyrosine (4PS)[20], is also activated and phosphorylated by IL-4 via the γC in macrophages. IL-4-induced Irs2 phosphorylation and M2a-subtype macrophage activation were decreased in γC-deficient macrophages[21]. However, the underlying mechanisms by which M2a-subtype macrophage activation by IL-4 was impaired in obesity remain poorly understood. To clarify whether Irs2 is involved in impaired IL-4-induced M2a-subtype macrophage activation, we analyzed myeloid cell-specific *Irs2*-deficient (M*Irs2*KO) mice under the high-fat (HF) diet condition, and attempted to determine the signaling mechanisms by which IL-4/Irs2 pathway would regulate M2a-subtype macrophage activation.

## Results

**M*Irs2*KO mice exhibited insulin resistance and inflammation**. To investigate the IL-4 signaling in MΦs in obesity, peritoneal MΦs were collected from normal chow (NC) and HF diet-fed mice after thioglycollate injection. The phosphorylation levels of STAT6 induced by IL-4 in the peritoneal MΦs did not differ between the NC and HF diet-fed mice (Fig. 1a). In contrast, the phosphorylation levels of Irs2 induced by IL-4 were significantly reduced along with decreased MΦ *Irs2* expression in the HF diet-fed mice (Fig. 1a and Supplementary Fig. 1a). SiglecF⁻CD11b⁺F4/80⁺ cells of the stromal vascular fraction (SVF) of the white adipose tissue (WAT) were collected from mice with genetically (ob/ob) or environmentally induced (15-week, HF diet-fed) obesity (Supplementary Fig. 1b). While there were no significant differences in the expression levels of *IR*, *IL-4R* or *STAT6* between the two mouse models of obesity (Supplementary Fig. 1b, c), the expression levels of *Irs2* mRNA were significantly reduced in both (Fig. 1b, c). Expression of Irs1 mRNA was undetectable in these cells, as previously reported (Supplementary Fig. 1c, d)[22]. Since it has been shown that the M1- and M2-type MΦ fractions are clearly separable using CD11c and CD206, the classic markers of M1- and M2-type MΦs, respectively[23], we investigated the expression levels of *Irs2* in siglecF⁻CD11b⁺F4/80⁺CD11c⁺ cells (M1-type MΦs) and siglecF⁻CD11b⁺F4/80⁺CD206⁺ cells (M2-type MΦs) in the SVCs of the adipose tissue derived from NC and HF diet-fed mice. The expression levels of *Irs2* mRNA were significantly reduced in both the types of cells derived from HF diet-fed mice (Supplementary Fig. 1e). These data suggest that IL-4/Irs2-mediated signaling in the MΦs is impaired in obesity.

To investigate the role of Irs2 in the MΦs, we then generated M*Irs2*KO mice. Although there were no significant differences in the mRNA expression levels of *IR* or *IL-4R* in the bone marrow-derived MΦs (BMDM) between the control and M*Irs2*KO mice, the *Irs2* mRNA and protein expressions were almost completely abrogated in the BMDM of the M*Irs2*KO mice (Supplementary Fig. 2a, b). Furthermore, *Irs1* mRNA expression was undetectable in the BMDM of both the control and M*Irs2*KO mice (Supplementary Fig. 2a). There were no significant differences in the body weight (BW), results of the insulin tolerance test (ITT) and oral glucose tolerance test (OGTT) or the lipid profiles between the two models of mice under the NC diet condition (Supplementary Fig. 2c-f). We analyzed the percentages of the siglecF⁻CD11b⁺F4/80⁺ cells, siglecF⁻CD11b⁺F4/80⁺CD11c⁺ cells and siglecF⁻CD11b⁺F4/80⁺CD206⁺ cells in the SVF of the adipose tissue derived from the NC-fed M*Irs2*KO mice. There were no differences in the percentages of any of these cells between the control and M*Irs2*KO mice under the NC diet condition (Supplementary Fig. 2g). Under HF diet condition, however, the blood glucose and plasma insulin levels during the OGTT were significantly higher in the M*Irs2*KO mice, although there were no significant differences in the BW gain, lean body mass or percent body fat (%FAT) between the control and M*Irs2*KO mice (Fig. 1d and Supplementary Fig. 3a, b). In the hyperinsulinemic-euglycemic clamp study, the M*Irs2*KO mice showed a reduced glucose infusion rate (GIR) and increased endogenous glucose production (EGP) (Fig. 1e). The phosphorylation levels of Akt in the liver and WAT, but not in the skeletal muscle, were significantly decreased in the M*Irs2*KO mice after insulin infusion via the vena cava (Fig. 1f). These data suggest that the lack of *Irs2* in the MΦs causes hepatic and WAT insulin resistance under the HF diet condition. In fact, the blood glucose levels during the pyruvate tolerance test (PTT) and expression levels of *phosphoenolpyruvate carboxykinase* (*PEPCK*) and *glucose 6-phosphatase* (*G6Pase*) in the liver were significantly increased in the HF diet-fed M*Irs2*KO mice (Supplementary Fig. 3c, d). Moreover, the hepatic triglyceride (TG) content was significantly increased in the HF diet-fed M*Irs2*KO mice (Supplementary Fig. 3e). There were no significant differences in the expression levels of *sterol regulatory element binding protein* (*SREBP1*)*c*,

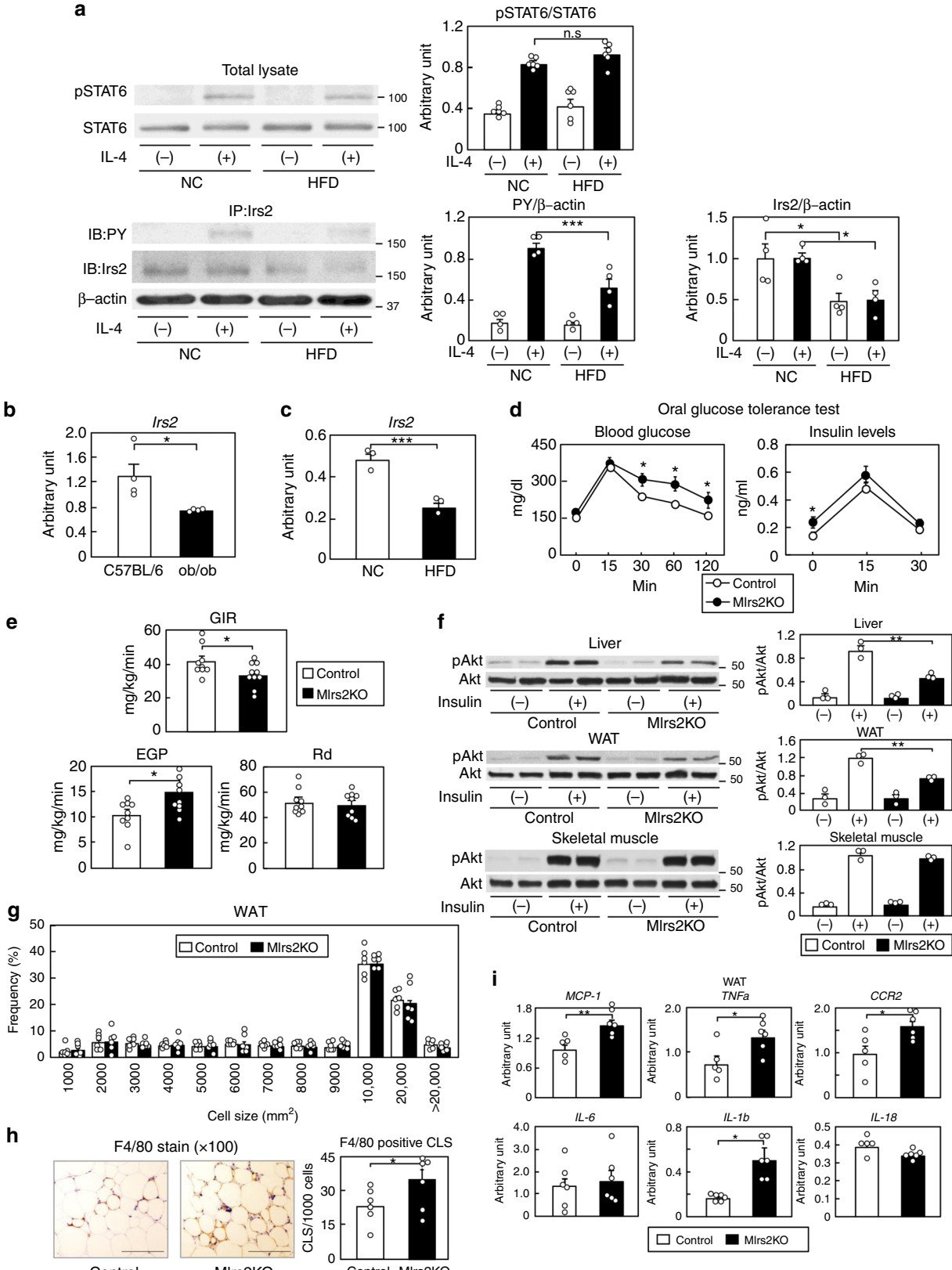

**Fig. 1** MΦ *Irs2* deficiency caused inflammation and insulin resistance in the liver and WAT under a HF diet. **a** Phosphorylation and protein levels of STAT6 and Irs2 in peritoneal MΦs from NC and HF diet-fed mice ($n = 4$). **b**, **c** Expression levels of *Irs2* mRNA in the siglecF-CD11b+F4/80+cells of the SVF of the adipose tissue from ob/ob and HF diet-fed mice ($n = 4$). **d** Glucose tolerance test in the M*Irs2*KO mice ($n = 7$–14). **e** GIR, EGP and Rd in the M*Irs2*KO mice in the hyperinsulinemic-euglycemic clamp study ($n = 9$–10). **f** Phosphorylation levels of Akt (ser473) in the liver, WAT and skeletal muscle of the M*Irs2*KO mice after insulin infusion ($n = 3$–4). **g** Adipocyte cell size in the M*Irs2*KO mice ($n = 6$). **h** F4/80-positive CLS in the WAT of M*Irs2*KO mice (scale bar, 200 μm) ($n = 6$). **i** Quantitative RT-PCR analysis of the genes encoding inflammatory cytokines in the WAT of the M*Irs2*KO mice ($n = 6$). The data are mean ± SEM. followed by one-way ANOVA with a post hoc test or Student's *t* test. *$P < 0.05$; **$P < 0.01$; ***$P < 0.001$

acetyl-CoA carboxylase (ACC), fatty acid synthase (FAS) or stearoyl-CoA desaturase (SCD)1 between the control and MIrs2KO mice, whereas the expression levels of PPARγ, fat-specific protein (FSP)27 and CD36 were significantly increased in the livers of the HF diet-fed MIrs2KO mice (Supplementary Fig. 3f). The expression levels of the inflammatory cytokines in the liver were also significantly elevated in the HF diet-fed MIrs2KO mice (Supplementary Fig. 3g). In the WAT, while the adipocyte cell size was similar between the control and MIrs2KO mice, F4/80-positive crown-like structures (CLS) were significantly increased in the HF diet-fed MIrs2KO mice (Fig. 1g, h). Moreover, the expression levels of the inflammatory cytokines were also significantly increased in the WAT obtained from these mice (Fig. 1i). These data suggest that a lack of Irs2 in the MΦs causes insulin resistance and inflammation in both the liver and the WAT under the HF diet condition.

**Irs2 deficiency impaired M2aΦ activation in obesity**. To investigate whether the polarization of MΦs is altered in the HF diet-fed MIrs2KO mice, the M1- and M2-type MΦs were analyzed by flow cytometry. Consistent with the results of the histological analyses (Fig. 1h), the percentage of siglecF⁻CD11b⁺F4/80⁺ cells was significantly increased in the WAT of the HF diet-fed MIrs2KO mice (Fig. 2a). Although the Irs2 expression levels in the siglecF⁻CD11b⁺F4/80⁺ cells of the MIrs2KO mice were markedly reduced, the expression levels of IR, IL-4R and STAT6 did not differ between the control and MIrs2KO mice under the HF diet condition (Supplementary Fig. 4a). The HF diet-fed MIrs2KO mice showed an increase in the number of CD11c-positive cells and decrease in the number of CD206-positive cells in the ATMs (Fig. 2a). The M1/M2 ratio was significantly increased in the adipose tissue of the HF diet-fed MIrs2KO mice (Fig. 2a). Consistent with these results, the M2a-subtype MΦ marker genes were downregulated, whereas some of the M1-type MΦ marker genes were upregulated in the siglecF⁻CD11b⁺F4/80⁺ cells of the MIrs2KO mice (Fig. 2b). The expression levels of IL-10, which is predominantly expressed in M2b-subtype MΦs, were not significantly different between the control and MIrs2KO mice. The percentage of siglecF⁺ cells in the SVCs were also not significantly different between the control and MIrs2KO mice (Supplementary Fig. 4b). These data suggest that the lack of Irs2 in the MΦs led to an increase in the number of M1-type MΦs and the M1/M2 ratio, and a decrease in the number of M2a-subtype MΦs in the WAT, resulting in the aggravation of insulin resistance in the MIrs2KO mice under the HF diet condition.

Why did the M1/M2 ratio increase in the WAT of the HF diet-fed MIrs2KO mice? Since M2a-subtype MΦs are activated by both IL-4 and IL-13[18], we investigated whether it is IL-4 or IL-13 that regulates the M2a-subtype MΦ marker gene expressions through Irs2. The expression levels of Arg1, FIZZ1, Ym1 and MΦ galactose N-acetyl-galactosamine-specific lectin 1 (Mgl1), which are hallmarks of activated M2a-subtype MΦs, were significantly reduced in the BMDM of the MIrs2KO mice after IL-4 stimulation (Fig. 2c). Consistent with these data, IL-4-induced arginase activity was also impaired in the BMDM of the MIrs2KO mice (Fig. 2d). In marked contrast, the expression levels of these marker genes after IL-13 stimulation were similar between the control and MIrs2KO mice (Supplementary Fig. 4c). There were no significant differences in the IL-4 and IL-13 levels in the WAT between the control and MIrs2KO mice under the HF diet condition (Supplementary Fig. 4d). These data suggest that IL-4-induced, but not IL-13-induced, M2a-subtype MΦ activation is impaired in the MIrs2KO mice. There were no significant differences in LPS-induced M1-type activation or MCP-1-induced BMDM migration between the control and MIrs2KO

mice (Fig. 2e and Supplementary Fig. 4e). Although IL-4 has been reported to regulate the proliferation of tissue-resident MΦs[24], IL-4-induced BMDM proliferation did not differ between the control and MIrs2KO mice (Supplementary Fig. 4f). We then constructed a co-culture system of BMDM derived from MIrs2KO mice and 3T3-L1 cells. The IL-4 levels were increased in the conditioned medium of the 3T3-L1 cells, but not BMDM (Supplementary Fig. 4g), suggesting that IL-4 was secreted from the 3T3-L1 cells. There was no significant difference in the IL-4 level in the conditioned medium between the BMDM of the control and MIrs2KO mice in co-culture (Supplementary Fig. 4g). The CCR2 and IL-6 expression levels in the 3T3-L1 cells were significantly higher than those in the BMDM from the MIrs2KO mice than in the BMDM derived from control mice (Fig. 2f). These data suggest that impairment of IL-4-induced M2a-subtype MΦ activation in the BMDM is responsible for the increased M1/M2 ratio in the HF diet-fed MIrs2KO mice.

We next investigated the molecule mechanisms of IL-4-induced M2a-subtype MΦ activation via Irs2. Although IL-4-induced phosphorylation of Irs2 in the BMDM of the control mice, this phosphorylation was undetectable in the BMDM of the MIrs2KO mice (Fig. 2g). Moreover, treatment with LY294002, a PI3 kinase inhibitor, significantly inhibited the phosphorylation of Akt induced by IL-4 in the BMDM (Fig. 2h). IL-4-induced Akt phosphorylation was significantly decreased in the BMDM of the MIrs2KO mice (Fig. 2i). Consistent with this result, the expression levels of Arg1, FIZZ1, Ym1 and Mgl1 induced by IL-4 were significantly suppressed by LY294002 treatment (Fig. 2j). We next examined whether FoxO1 phosphorylation is regulated by IL-4 in the BMDM. FoxO1 was phosphorylated by IL-4, and this phosphorylation was inhibited by LY294002 treatment (Supplementary Fig. 4h). Moreover, immunohistochemical staining revealed that FoxO1 was translocated from the nucleus to the cytoplasm after IL-4 stimulation, and that this translocation was inhibited by LY294002 treatment (Supplementary Fig. 4i). Consistent with these data, we found that after IL-4 stimulation, the FoxO1 protein levels were decreased in the nuclear fraction, whereas they were increased in the cytoplasmic fraction (Supplementary Fig. 4j). We then investigated whether FoxO1 is involved in M2a-subtype MΦ activation. Transfection of constitutively active (CA) FoxO1 significantly reduced the expression levels of Arg1, FIZZ1, Ym1 and Mgl1 induced by IL-4 stimulation (Fig. 2k). IL-4-induced FoxO1 phosphorylation was significantly reduced in the BMDM of the MIrs2KO mice as compared to the control mice (Fig. 2l), and the decreased expression levels of the M2a-subtype marker genes in the BMDM of the MIrs2KO mice were completely restored by siFoxO1 treatment (Fig. 2m). The IL-10 and TNFα expression levels were significantly decreased by IL-4 stimulation and their gene expression levels did not differ among the three groups. Although the IL-1β expression levels were also significantly decreased by IL-4 stimulation in all three groups, the transfection of siFoxO1 reduced the expression levels of IL-1β before IL-4 stimulation in the BMDM of the MIrs2KO mice. No IL-12a or IL-12b expression was detected in the BMDM (Supplementary Fig. 4k). These data suggest that IL-4 regulates the PI3 kinase-FoxO1 pathway via Irs2, which is essential for the activation of M2a-subtype MΦs.

**The FoxO1/NCoR1/HDAC3 complex suppressed M2aΦ activation**. To explore the mechanisms underlying FoxO1 mediation of M2a-subtype MΦ activation induced by IL-4 treatment, we measured the Arg1 promoter activity with or without CA-FoxO1 treatment using RAW264.7 cells. We first confirmed that the Irs2 protein was expressed, and then that IL-4-induced M2a-subtype MΦ activation and that this activation was inhibited by LY294002

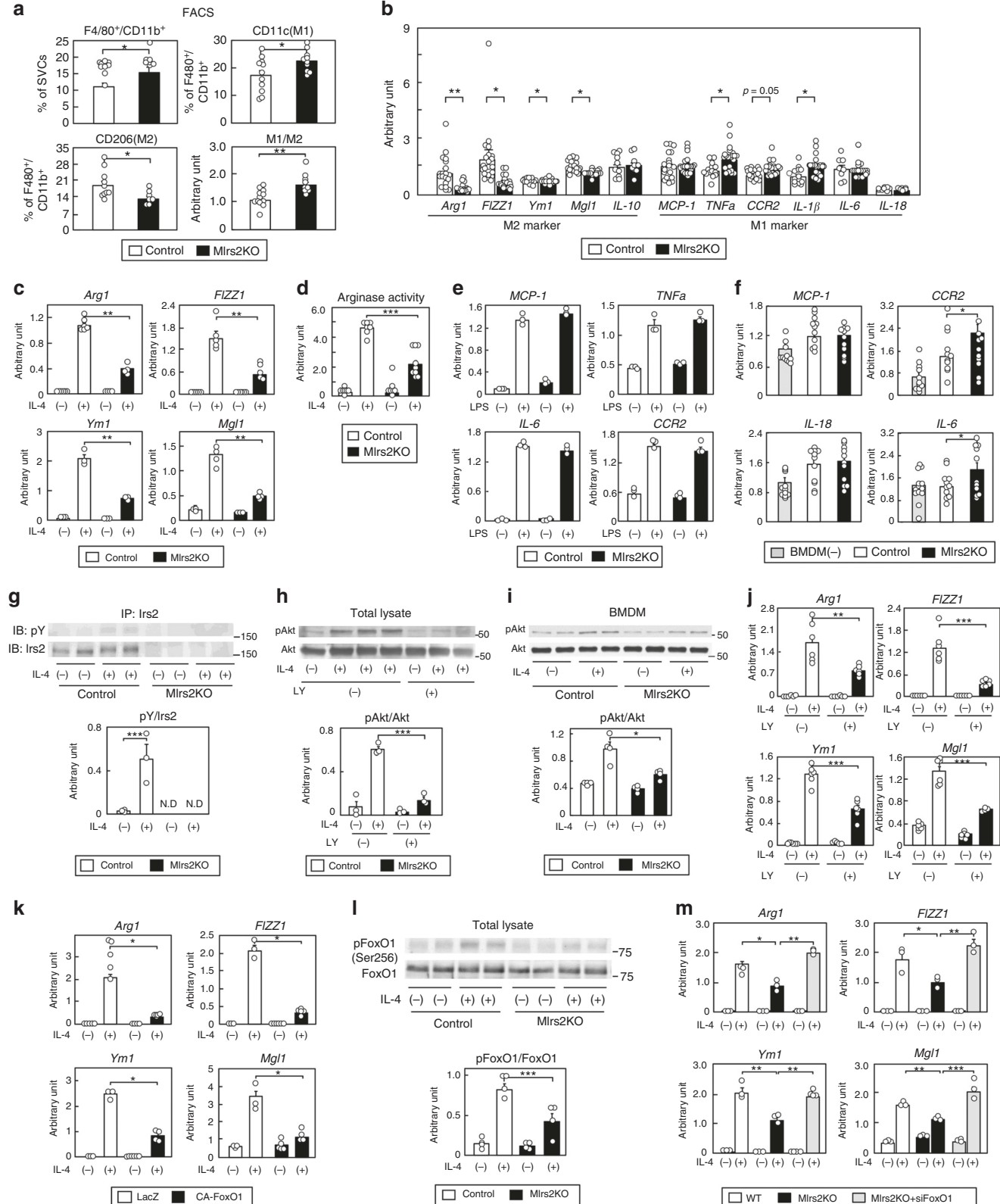

treatment in the RAW264.7 cells (Supplementary Fig. 5a, b). We created *Arg1* luciferase constructs including enhancer elements and proximal promoter fragments on the basis of a previous report[25]. The enhancer elements contained the STAT6 response element, which is considered as a STAT6-binding site. Although the *Arg1* promoter activity was significantly increased after IL-4 stimulation, this increase was suppressed by CA-FoxO1 treatment

(Fig. 3a). These data suggest that FoxO1 directly regulates IL-4-induced *Arg1* transcript activation. Since we found two FoxO1 consensus sequences in the promoter area of the *Arg1* constructs, we created an *Arg1* construct with either A (GT<u>AAATAA</u> → GT<u>CGATAA</u>) or B (<u>AAAACAA</u> → <u>TCTAGACA</u>) area mutation. Mutant A promoter activity was significantly increased after IL-4 stimulation, and this increase was suppressed by CA-FoxO1

**Fig. 2** IL-4-induced M2a-subtype MΦ activation was impaired in WAT of the HF diet-fed M*Irs2*KO mice. **a** The percentages of siglecF⁻CD11b⁺F4/80⁺, siglecF⁻CD11b⁺F4/80⁺CD11c⁺CD206⁻ (M1-type MΦs) and siglecF⁻CD11b⁺F4/80⁺CD11c⁻CD206⁺ (M2-type MΦs) cells in the SVF of the adipose tissue ($n = 10$–14). **b** Expression levels of M2-type and M1-type MΦ marker genes in the siglecF⁻CD11b⁺F4/80⁺ cells in the SVF of the adipose tissue from the control and M*Irs2*KO mice ($n = 10$–14). **c** Expression levels of M2a-subtype MΦ marker genes in the BMDM of the control and M*Irs2*KO mice after IL-4 stimulation for 48 h ($n = 6$–8). **d** Arginase activity in the BMDM of the control and M*Irs2*KO mice after IL-4 stimulation for 24 h ($n = 7$–10). **e** Expression levels of the M1-type MΦ marker genes in the BMDM of the control and M*Irs2*KO mice after LPS stimulation for 48 h ($n = 3$). **f** MCP-1, CCR2, IL-18 and IL-6 expression levels in 3T3-L1 cells in co-culture with the BMDM of M*Irs2*KO mice ($n = 12$). **g** Irs2 phosphorylation and protein levels in the BMDM of the control and M*Irs2*KO mice after IL-4 stimulation ($n = 3$–4). **h** Akt phosphorylation and protein levels in the BMDM of the C57BL/6 mice after IL-4 stimulation with or without LY294002 treatment ($n = 3$). **i** IL-4-induced Akt phosphorylation in the BMDM of the M*Irs2*KO mice ($n = 4$). **j** Expression levels of the M2a-subtype MΦ marker genes in the BMDM of the C57BL/6 mice after IL-4 stimulation with or without LY294002 treatment ($n = 4$–6). **k** Expression levels of the M2a-subtype MΦ marker genes in the BMDM of the C57BL/6 mice after IL-4 stimulation with or without CA-FoxO1 treatment ($n = 4$–6). **l** FoxO1 phosphorylation and protein levels in the BMDM of the control and M*Irs2*KO mice after IL-4 stimulation ($n = 3$–4). **m** Expression levels of the M2a-subtype MΦ marker genes in the BMDM of the M*Irs2*KO mice after IL-4 stimulation with siFoxO1 transfection ($n = 5$–6). The data are mean ± SEM. followed by one-way ANOVA with a post hoc test or Student's $t$ test. *$P < 0.05$; **$P < 0.01$; ***$P < 0.001$

treatment (Fig. 3a, b; left panel); on the other hand, mutant B and mutant A plus B promoter activities were not inhibited by CA-FoxO1 treatment (Fig. 3a, b; middle and right panel). These data suggest that FoxO1 directly binds to consensus sites including the B area, and suppresses *Arg1* transcript activation induced by IL-4. Since FoxO1 is known to form a corepressor complex with HDAC and NCoR1 in the central nervous system[26], we investigated whether FoxO1 forms a corepressor complex with HDAC3 and NCoR1 in the RAW264.7 cells by using the electrophoretic mobility shift assay (EMSA). The EMSA revealed that DNA nucleotides of the B area in the *Arg1* promoter bind to proteins, and that the DNA-protein complexes were blocked or gel-shifted in a dose-dependent manner by FoxO1, HDAC3 or NCoR1 antibody (Fig. 3c). Moreover, we performed Chip-qPCR using FoxO1, HDAC3 or NCoR1 antibody in the Raw 264.7 cells and BMDM. Although the *Arg1* expression levels were significantly increased before IL-4 stimulation, these increases were inhibited after IL-4 stimulation in both the Raw 264.7 cells and BMDM (Fig. 3d and Supplementary Fig. 5c). These data suggest that FoxO1 forms a corepressor complex with HDAC3 and NCoR1, and that this corepressor complex suppresses *Arg1* transcript activation. Moreover, we performed the Chip-qPCR assay in the BMDM of the M*Irs2*KO mice. Although the *Arg1* expression levels were significantly increased before IL-4 stimulation, these increases were not inhibited after IL-4 stimulation in the BMDM of the M*Irs2*KO mice (Fig. 3e). There were no significant differences in the *FoxO1*, *HDAC3* or *NCoR1* expression levels in the ATMs between the control and M*Irs2*KO mice (Supplementary Fig. 5d). These data suggest that Irs2 is essential for IL-4-induced dissociation of the FoxO1/HDAC3/NCoR1 corepressor complex in vivo. To investigate whether FoxO1 directly binds to HDAC3 and NCoR1, we conducted co-immunoprecipitation (Co-IP) with FoxO1, HDAC3, or NCoR1 antibody without IL-4 stimulation in the Raw 264.7 cells and BMDM. Co-IP with FoxO1 detected HDAC3 and NCoR1, Co-IP with HDAC3 detected FoxO1 and NCoR1, and Co-IP with NCoR1 detected FoxO1 and HDAC3 in both the Raw 264.7 cells and BMDM (Fig. 3f). These data suggest that FoxO1 directly binds to HDAC3 and NCoR1. Moreover, IL-4-induced *Arg1* expression was significantly increased in the BMDM transfected with siHDAC3 or siNCoR1 (Fig. 3g). In addition to *Arg1*, the *FIZZ1* and *Ym1* promoter activities were also significantly increased after IL-4 stimulation, although these increases were suppressed by CA-FoxO1 (Supplementary Fig. 5e). IL-4-induced *FIZZ1* and *Ym1* expressions were significantly increased in the BMDM transfected with siHDAC3 or siNCoR1 (Supplementary Fig. 5f). The *TNFα* and *IL-1β* expression levels were significantly decreased by IL-4 stimulation, while their gene expression levels did not differ between the BMDM transfected and not transfected with siHDAC3 or siNCoR1. No *IL-12a* or *IL-*

*12b* expression was detected in the BMDM (Supplementary Fig. 5g). These data suggest that FoxO1 directly binds to HDAC3 and NCoR1 and forms a corepressor complex, leading to suppression of M2a-subtype MΦ activation.

**M*IR*KO mice exhibit improved insulin sensitivity**. Irs2 is known as a substrate for not only IL-4R, but also IR. Myeloid cell-specific IL-4Rα-deficient (M*IL-4R*KO) mice showed decreased M2a-subtype MΦ activation and insulin sensitivity, just like the M*Irs2*KO mice[16]. Thus, we next generated and investigated the phenotypes of M*IR*KO mice. As compared to the M*Irs2*KO mice, the glucose tolerance was significantly better and the plasma insulin levels significantly lower in the M*IR*KO mice, although there was no significant difference in the BW gain between the two models of mice (Fig. 4a and Supplementary Fig. 6a). The M*IR*KO mice showed increased GIR and reduced EGP in the hyperinsulinemic-euglycemic clamp study (Fig. 4b). The phosphorylation levels of Akt in the liver and WAT, but not in the skeletal muscle, were significantly increased in the M*IR*KO mice after insulin infusion via the vena cava (Fig. 4c). These data suggest that the M*IR*KO mice show an improvement of the hepatic and WAT insulin resistance under the HF diet condition. Consistent with these data, PTT revealed that the blood glucose levels were lower in the M*IR*KO mice than in the control mice, and that *PEPCK* and *G6Pase* expression levels were significantly reduced in the livers of the M*IR*KO mice (Supplementary Fig. 6b, c). The TG content and gene expression levels associated with lipogenesis were significantly decreased in the livers of the M*IR*KO mice (Supplementary Fig. 6d, e). Moreover, the M*IR*KO mice also showed a reduction in the expression levels of the inflammatory cytokines in the liver (Supplementary Fig. 6f). In the WAT of the HF diet-fed M*IR*KO mice, the number of F4/80-positive CLS was significantly decreased, although there was no significant change of the adipocyte cell size (Fig. 4d, e). The HF diet-fed M*IR*KO mice showed decreased inflammatory cytokine expression levels in the WAT (Fig. 4f). Moreover, the number of CD206-positive cells was increased, while the proportions of siglecF⁻CD11b⁺F4/80⁺ cells, CD11c-positive cells and the M1/M2 ratio were decreased in the WAT of the HF diet-fed M*IR*KO mice (Fig. 4g). Consistent with these results, the M2a-subtype MΦ maker genes were upregulated, whereas some of M1-type MΦ marker genes were downregulated in the siglecF⁻CD11b⁺F4/80⁺ cells of the HF diet-fed M*IR*KO mice (Fig. 4h). The expression levels of *IL-10* were not significantly different between the control and M*IR*KO mice. The percentage of siglecF⁺ cells in the SVCs, and the IL-4 and IL-13 levels in the WAT did not differ between the control and M*IR*KO mice under the HF diet condition (Supplementary Fig. 6g, h). Although the *IL-4R* and *Irs2* mRNA expression levels in the BMDM did not differ between the control

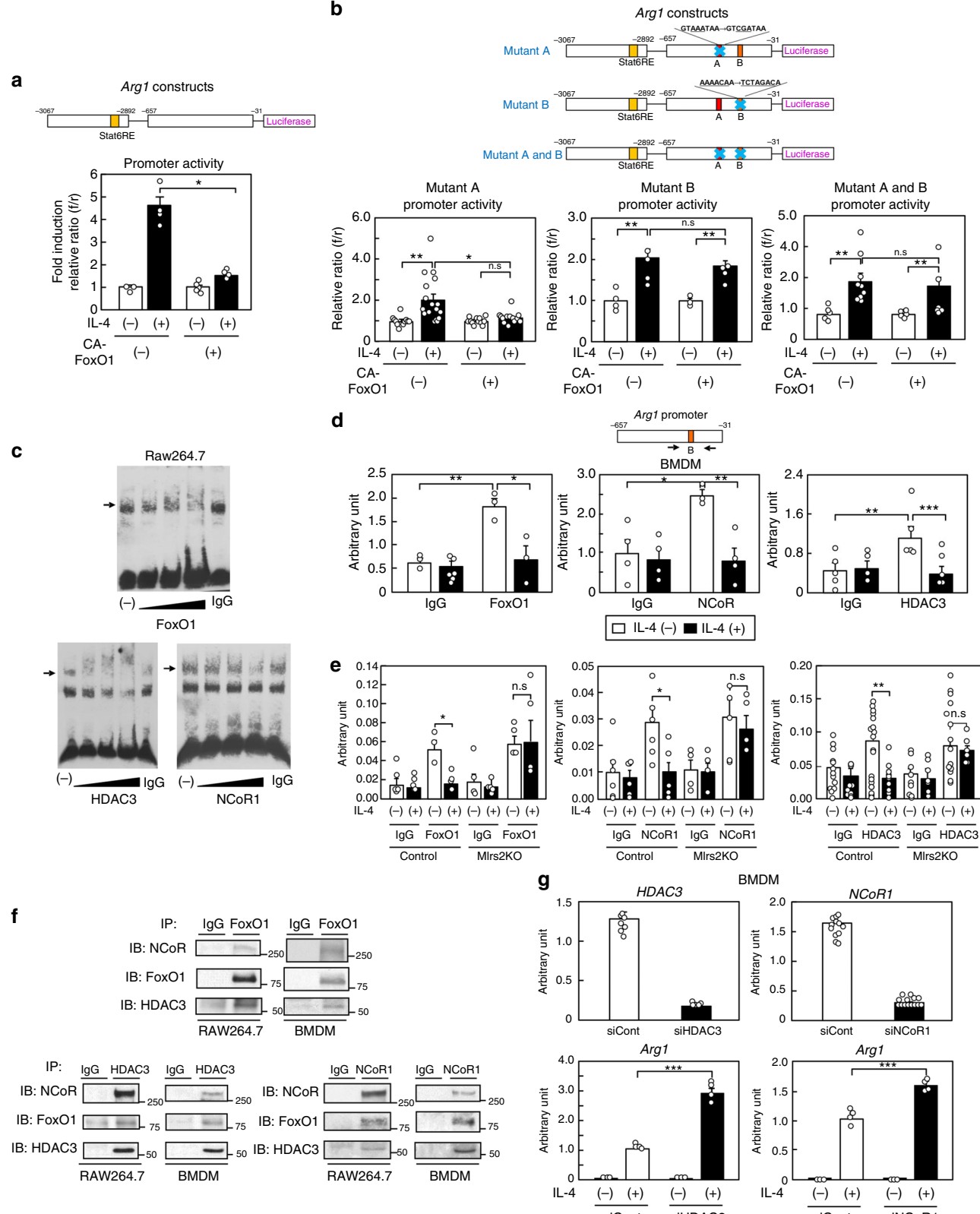

**Fig. 3** FoxO1 bound to HDAC3 and NCoR1 to form a corepressor complex. **a** *Arg1* promoter activity after IL-4 stimulation with or without CA-FoxO1 transfection ($n = 4$–5). **b** Mutant A, B, and A plus B *Arg1* promoter activities after IL-4 stimulation with or without CA-FoxO1 transfection ($n = 5$–6). **c** The binding of DNA nucleotides of the B area of the *Arg1* promoter and proteins, and the blocking of the DNA-protein complex formation in a dose-dependent manner by FoxO1, HDAC3 and NCoR1 antibody were determined by EMSA in RAW264.7 cells. **d** Chip-qPCR using FoxO1, HDAC3 and NCoR1 antibody in the BMDM of the C57BL/6 mice before and after IL-4 stimulation ($n = 5$–10). **e** Chip-qPCR using FoxO1, HDAC3 and NCoR1 antibody in the BMDM of the M*Irs2*KO mice before and after IL-4 stimulation ($n = 3$–16). **f** Co-immunoprecipitation (Co-IP) with FoxO1, HDAC3 and NCoR1 antibody without IL-4 stimulation in the Raw 264.7 cells and BMDM. **g** Expression levels of *Arg1* in the BMDM of the C57BL/6 mice after IL-4 stimulation with siHDAC3 or siNCoR1 transfection ($n = 3$–8). The data are mean ± SEM. followed by one-way ANOVA with a post hoc test. *$P < 0.05$; **$P < 0.01$; ***$P < 0.001$

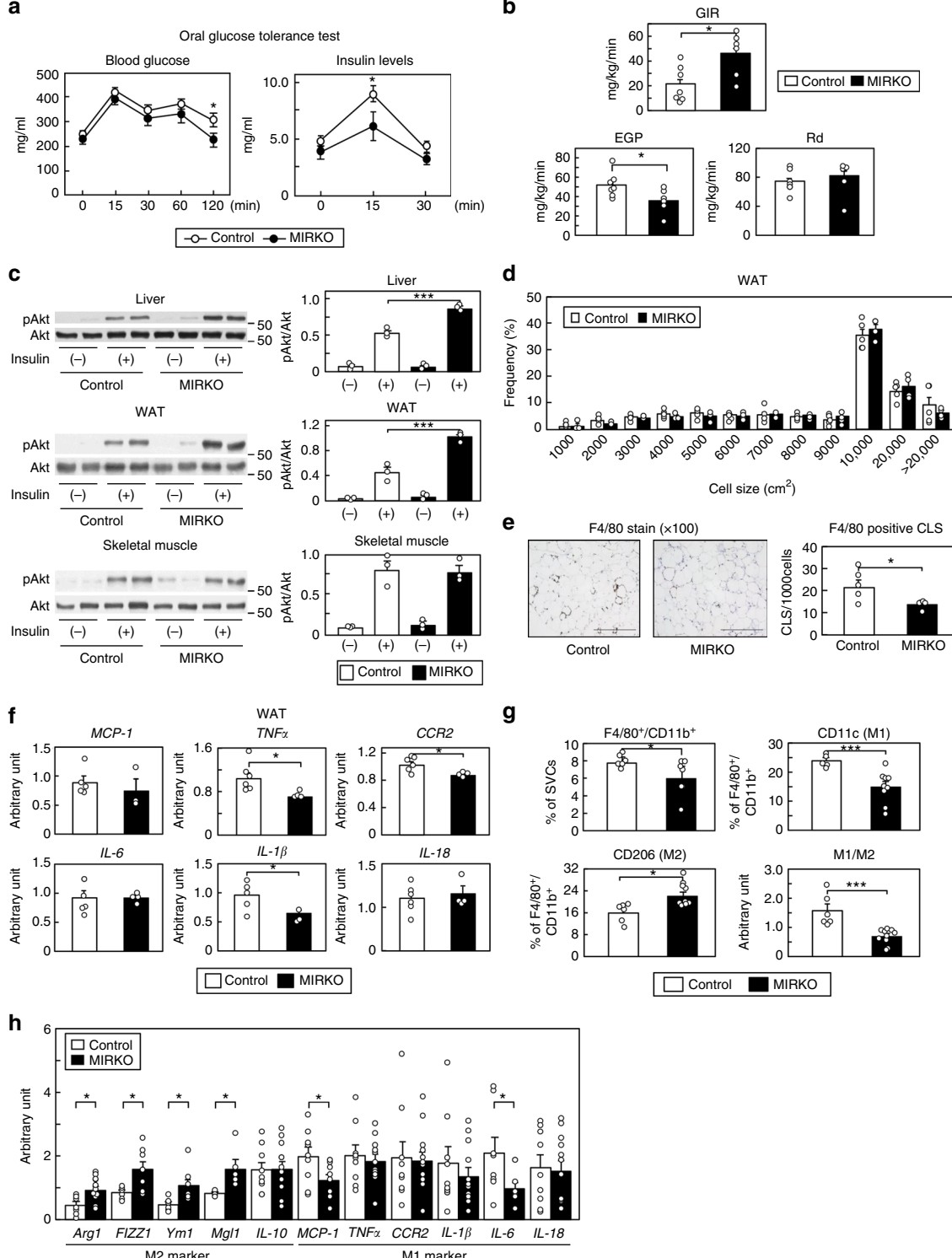

**Fig. 4** M*IR*KO mice exhibited improved insulin sensitivity and glucose tolerance. **a** Glucose tolerance test in the M*IR*KO mice ($n = 7–13$). **b** GIR, EGP and Rd in the M*IR*KO mice in the hyperinsulinemic-euglycemic clamp study ($n = 8–9$). **c** Phosphorylation levels of Akt (ser473) in the liver, WAT and skeletal muscle of the M*IR*KO mice after insulin infusion ($n = 3–4$). **d** Adipocyte cell size in the M*IR*KO mice ($n = 4–6$). **e** F4/80-positive CLS in the WAT of the M*IR*KO mice (scale bar, 200 μm) ($n = 4–6$). **f** Quantitative RT-PCR analysis of the genes encoding inflammatory cytokines in the WAT of the M*IR*KO mice ($n = 4–6$). **g** The percentages of siglecF⁻CD11b⁺F4/80⁺, siglecF⁻CD11b⁺F4/80⁺CD11c⁺CD206⁻ (M1-type MΦs) and siglecF⁻CD11b⁺F4/80⁺CD11c⁻CD206⁺ (M2-type MΦs) cells in the SVF of the adipose tissue from the M*IR*KO mice ($n = 4–6$). **h** Expression levels of M2-type and M1-type MΦ marker genes in the siglecF⁻CD11b⁺F4/80⁺ cells in the SVF of the adipose tissue from the control and M*IR*KO mice ($n = 4–5$). Data are mean ± SEM. followed by one-way ANOVA with a post hoc test or Student's *t* test. *$P < 0.05$; ***$P < 0.001$

and M*IR*KO mice under the NC diet condition, the *IR* mRNA and protein expressions were almost completely abrogated in the BMDM of the M*IR*KO mice (Supplementary Fig. 6i, j). There were no significant differences in the BW, ITT, GTT or lipid profiles between the control and M*IR*KO mice under the NC diet condition (Supplementary Fig. 6k-n). These data suggest that the lack of *IR* in the MΦs led to a decrease in the number of M1-type MΦs and the M1/M2 ratio, and an increase in the number of M2a-subtype MΦs in the WAT, resulting in the protection of insulin resistance in the M*IR*KO mice under the HF diet condition.

**Irs2 expressions and M2aΦ activation increased in M*IR*KO mice**. Unlike the M*Irs2*KO mice, why did the M1/M2 ratio decrease in the WAT of the HF diet-fed M*IR*KO mice? We analyzed the expression levels of *IR*, *Irs2*, *IL-4R*, *STAT6* in the ATMs of the M*IR*KO mice. Although the *IL-4R* and *STAT6* expression levels were not significantly different, the expression levels of *Irs2* were significantly higher in the siglecF⁻CD11b⁺F4/80⁺ cells isolated from the WAT of the M*IR*KO mice (Fig. 5a). Moreover, the IL-4-induced Irs2 and Akt phosphorylation were significantly enhanced in the peritoneal MΦs of HF diet-fed M*IR*KO mice, along with increased MΦ *Irs2* expression, although IL-4-induced STAT6 phosphorylation did not differ between control and M*IR*KO mice (Fig. 5b and Supplementary Fig. 7a). These data suggest that M2a-subtype MΦ activation was caused by enhanced IL-4-induced Irs2/Akt pathway along with increased *Irs2* expression levels in the M*IR*KO mice reared on a HF diet. IR signaling has been shown to suppress the expression of *Irs2*, both in vitro and in vivo, by inhibiting the synthesis of *Irs2* mRNA at the transcriptional level in organs such as the liver and endothelial cells[27–30]. Thus, we investigated the contribution of IR signaling to the expression levels of *Irs2* in the MΦs. Transfection of siFoxO1 reduced the *Irs2* expression in BMDM (Supplementary Fig. 7b). Although insulin stimulation markedly suppressed the expression of *Irs2* along with translocation of FoxO1 from the nucleus to the cytosol in the BMDM of the control mice (Supplementary Fig. 7c), this downregulation was not observed in the BMDM from the M*IR*KO mice (Fig. 5c). These data suggest that M2a-subtype MΦ activation via the IL-4/Irs2 pathway is maintained in M*IR*KO mice, resulting in reduced inflammation and insulin resistance, as *Irs2* expression in the MΦs of the M*IR*KO mice is not downregulated by insulin. We next investigated the expression levels of *Irs2* in the peritoneal MΦs of the mice after streptozotocin (STZ) plus phlorizin treatment. While the plasma insulin levels were significantly reduced, the peritoneal MΦ *Irs2* expression levels were significantly increased in the STZ plus phlorizin-treated mice (Supplementary Fig. 7d), suggesting that hyperinsulinemia downregulates MΦ *Irs2* expression in vivo. Downregulation of *Irs2* expression levels by pretreatment with insulin was associated with significantly reduced expressions of IL-4-induced M2a-subtype marker genes in the BMDM, while having no effect on the expression levels of *IR* and *IL-4R* (Fig. 5d and Supplementary Fig. 7e). The decreased IL-4-induced *Arg1* expression in the BMDM after insulin pretreatment was completely restored by siFoxO1, siNCoR1, or siHDAC3 treatment (Fig. 5e). Moreover, when the BMDM were co-cultured with 3T3-L1 cells in the presence of insulin, significant downregulation of the *Irs2* expression in the BMDM and significant upregulation of *MCP-1*, *CCR2*, *IL-18* and *IL-6* expressions in the 3T3-L1 cells were observed (Fig. 5f and Supplementary Fig. 7f, g).

**M2aΦ activation is regulated by both Irs2 and STAT6 pathways**. Why did insulin fail to induce M2a-subtype MΦ activation, although it could activate Irs2 as well as IL-4? Although insulin induced phosphorylation of Akt to the same level as that induced

by IL-4, it failed to induce M2a-subtype MΦ activation (Fig. 6a, b). These data suggest that IL-4-induced M2a-subtype MΦ activation is not mediated by the Irs2/Akt/FoxO1 pathway alone. Since IL-4 is well known to regulate M2a-subtype MΦ activation via the STAT6 pathway[18], we investigated STAT6 phosphorylation induced by IL-4 or insulin stimulation. While IL-4-induced phosphorylation of STAT6, insulin failed to do so (Fig. 6c). As in the case of the BMDM lacking *Irs2*, IL-4-induced M2a-subtype activation was reduced in the BMDM transfected with siSTAT6 (Fig. 6d). IL-4-induced STAT6 phosphorylation in the BMDM did not differ between the control and M*Irs2*KO mice, but this phosphorylation was not induced by insulin in the BMDM (Fig. 6e). Moreover, LY294002 treatment inhibited IL-4-induced Akt phosphorylation, but had no effect on IL-4-induced STAT6 phosphorylation (Figs. 2h, 6f). Consistent with these data, no differences were found in the expression levels of *PGC1β* or *PPARγ*, which are STAT6-associated genes, between the control and M*Irs2*KO mice after IL-4 treatment (Supplementary Fig. 8). These data suggest that IL-4 regulates M2a-subtype MΦ activation through both the STAT6 signaling and Irs2/Akt signaling pathways. In contrast, insulin rather suppressed IL-4-induced M2a-subtype MΦ activation via downregulation of *Irs2*.

**Discussion**

In this study, we demonstrated that IL-4/Irs2/Akt pathway was essential for activation of M2a-subtype MΦs, in addition to IL-4/STAT6 pathway. Activation of both pathways was necessary for the full activation of M2a-subtype MΦs (Fig. 7, left panel). In obesity, although IL-4/STAT6 pathway was preserved, IL-4/Irs2/Akt pathway was selectively impaired due to downregulation of *Irs2* induced by hyperinsulinemia via IR signaling, leading to impaired signaling in the IL-4/Irs2/Akt pathway and M2a-subtype MΦ activation by stabilization of the FoxO1/HDAC3/NCoR1 corepressor complex (Fig. 7, right panel). These data suggest that in obesity, hyperinsulinemia plays crucial roles in the dysregulation of M2a-subtype MΦ activation due to downregulation of *Irs2* expression.

The chronic hyperinsulinemia induced by sustained overnutrition, such as in animals reared on a HF diet, promotes adipocyte hypertrophy and obesity. In fact, adipose tissue-specific *IR*KO mice with gold thioglucose (GTG)-induced obesity exhibited a smaller adipocyte size in spite of hyperinsulinemia[31]. Enlarged adipocytes in response to hyperinsulinemia release inflammatory cytokines such as *MCP-1* and *TNFα*, leading to inflammatory monocytic infiltration of the WAT[32]. The infiltrating monocytes differentiate into M1-type MΦs and form CLS around dead adipocytes. The M1-type MΦs in the CLS also express inflammatory cytokines, thereby driving the development of insulin resistance. In fact, M1-type MΦs were recruited in mice showing overexpression of *MCP-1* in the adipose tissue, resulting in the development of insulin resistance[33]. On the other hand, in this study, we found that hyperinsulinemia downregulated *Irs2* expression, which led to impaired M2a-subtype MΦ activation. These data suggest that chronic hyperinsulinemia causes both increased M1-type MΦ activation via inducing adipocyte hypertrophy and decreased M2a-subtype MΦ activation via *Irs2* downregulation, thereby leading to the development of inflammation and insulin resistance.

Previous studies have demonstrated that M*IR*KO mice are protected from obesity-induced inflammation and insulin resistance under the HF diet condition. The obesity-associated MΦ infiltration of the adipose tissue was significantly reduced in these mice[34]. On the other hand, mice lacking *3-phosphoinositide-dependent protein kinase 1 (PDK1)* in the myeloid cells, which are

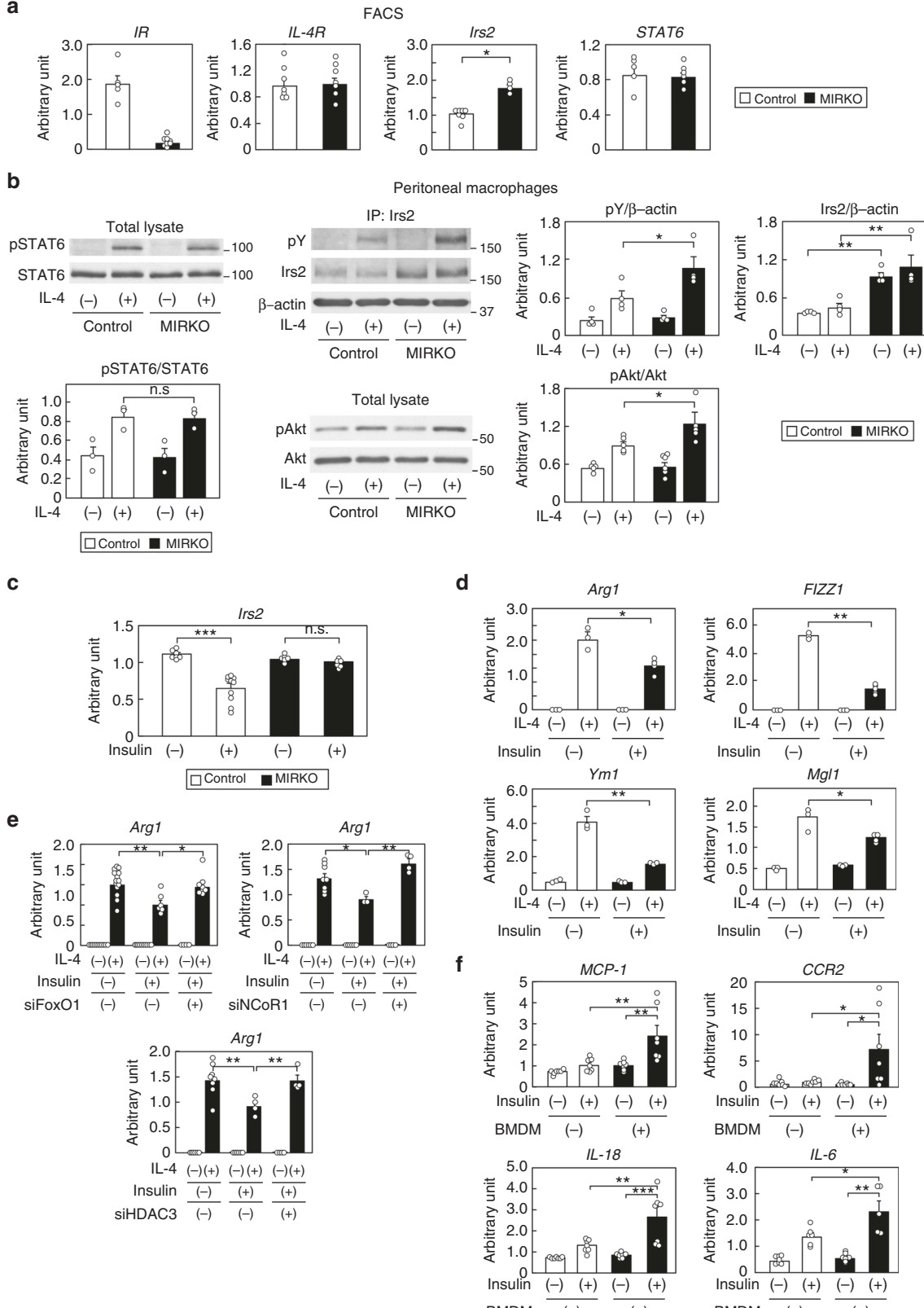

**Fig. 5** The *Irs2* mRNA levels and number of M2a-subtype MΦs were elevated in HF diet-fed M*IR*KO mice. **a** Expression levels of *IR*, *IL-4R*, *Irs2* and *STAT6* in the siglecF⁻CD11b⁺F4/80⁺ cells of the SVF of the adipose tissue from the control and M*IR*KO mice (*n* = 4–6). **b** IL-4-induced STAT6, Irs2 and Akt phosphorylation in peritoneal MΦs of HF diet-fed M*IR*KO mice (*n* = 3–6). **c** *Irs2* expression levels in the BMDM of the control and M*IR*KO mice after 100 nM insulin stimulation for 3 h (*n* = 3–4). **d** IL-4-induced M2a-subtype marker genes in the BMDM after 100 nM insulin pretreatment for 8 h (*n* = 3–4). **e** IL-4-induced *Arg1* expression levels in the BMDM pretreated with 100 nM insulin for 8 h after siFoxO1, siNCoR1, or siHDAC3 treatment (*n* = 4–16). **f** *MCP-1*, *CCR2*, *IL-18* and *IL-6* expression levels in 3T3-L1 cells in co-culture with BMDM of the C57BL/6 mice and 3T3-L1 cells after insulin stimulation for 24 h (*n* = 7). The data are mean ± SEM. followed by one-way ANOVA with a *post hoc* test or Student's *t* test. **P* < 0.05; ***P* < 0.01; ****P* < 0.001

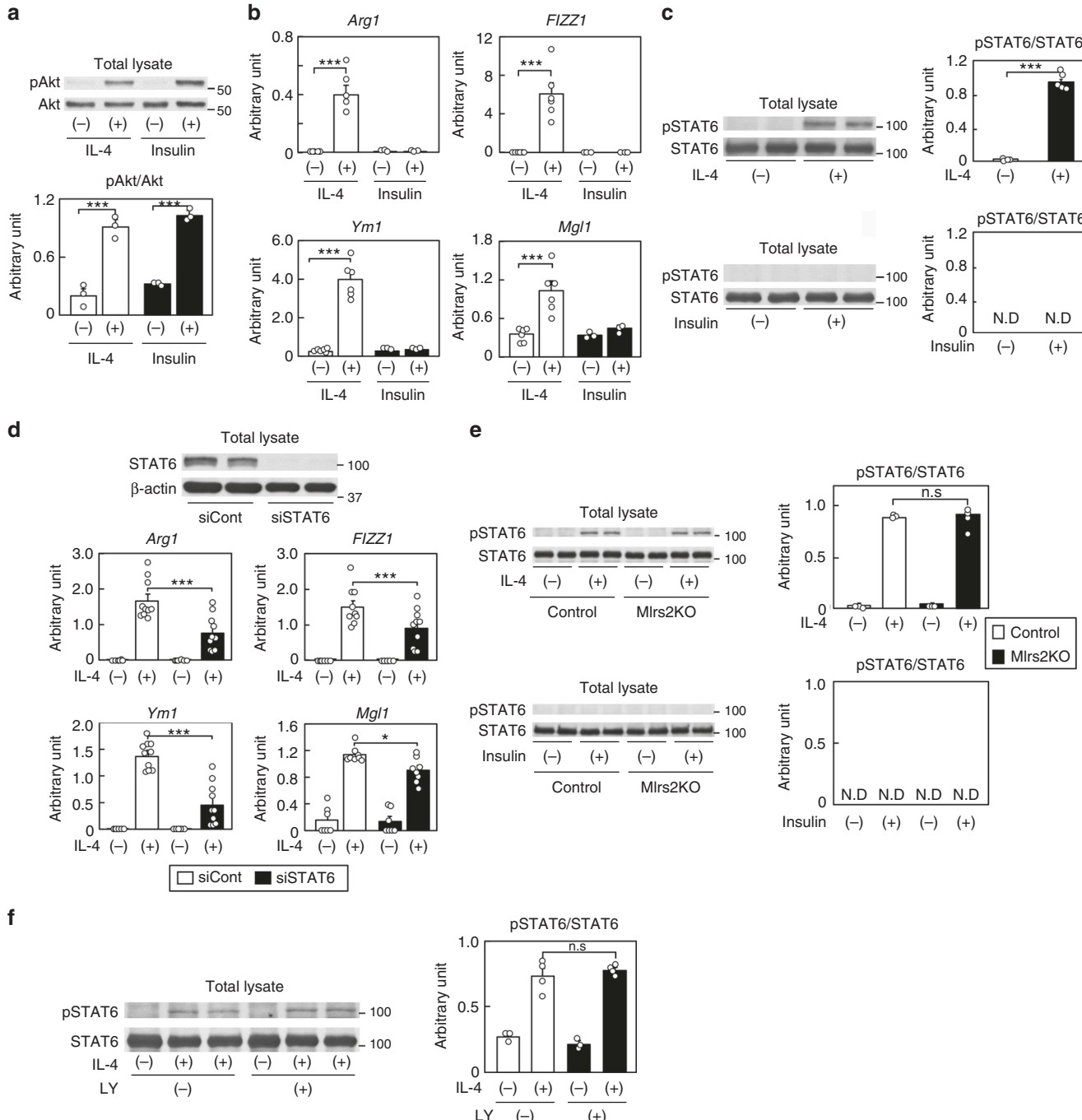

**Fig. 6** IL-4-induced M2a-subtype MΦ activation was regulated by both the Irs2 and STAT6 pathways. **a** Akt phosphorylation and protein levels in the BMDM of the C57BL/6 mice after IL-4 or insulin stimulation ($n = 3$–4). **b** Expression levels of the M2a-subtype MΦ marker genes in the BMDM of the C57BL/6 mice after IL-4 or insulin stimulation ($n = 4$). **c** STAT6 phosphorylation and protein levels in the BMDM of the C57BL/6 mice after IL-4 or insulin stimulation ($n = 3$–4). **d** Expression levels of the M2a-subtype MΦ marker genes in the BMDM of the C57BL/6 mice after IL-4 stimulation following siSTAT6 transfection ($n = 5$–6). **e** STAT6 phosphorylation and protein levels in the BMDM of the control and M*Irs2*KO mice after IL-4 or insulin stimulation ($n = 3$–4). **f** STAT6 phosphorylation and protein levels in the BMDM of the C57BL/6 mice after IL-4 stimulation following LY294002 treatment ($n = 4$). N. D not-detected. The data are mean ± SEM. followed by one-way ANOVA with a post hoc test. $^*P < 0.05$; $^{***}P < 0.001$

downstream signaling molecules in the IR/Irs2 pathway, showed adipose tissue inflammation and insulin resistance under the HF diet condition[35]. The results from the M*IR*KO mice suggest that impaired insulin signaling in MΦs improved obesity-induced inflammation and insulin resistance. On the other hand, impaired insulin signaling in the MΦs of *PDK1*KO mice aggravated inflammation and insulin resistance. Thus, in terms of the effects of insulin signaling, there seem to be discrepancies in the

phenotypes between the M*IR*KO and *PDK1*KO mice. However, this could be interpreted by analyzing the effects of IL-4 signaling in an integrated fashion. As seen in this study (Fig. 7), in the M*IR*KO mice, MΦ *Irs2* downregulation by insulin is suppressed and M2a-subtype MΦ activation via the IL-4 signaling pathway is preserved. Although MΦ *Irs2* downregulation by insulin may also be suppressed in the *PDK1*KO mice, as in the M*IR*KO mice, the IL-4 signaling pathway in the MΦs is impaired downstream of *Irs2*

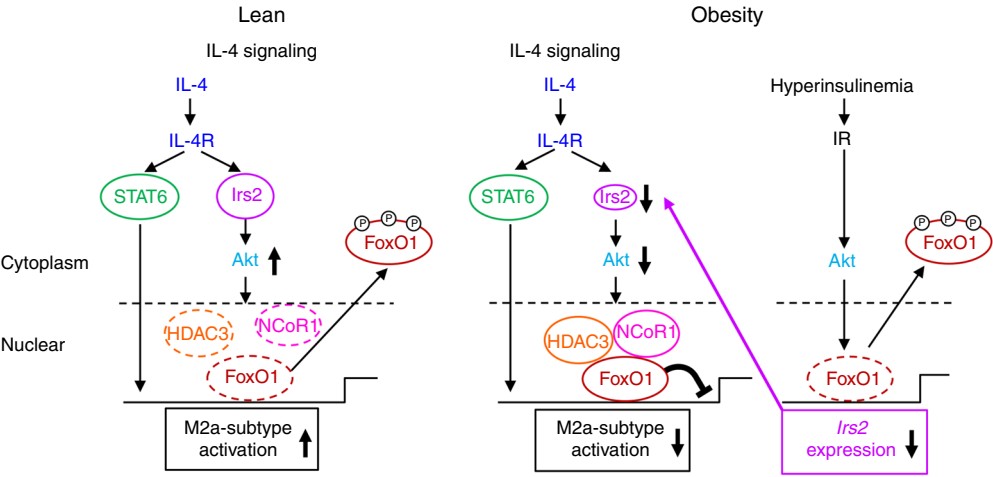

**Fig. 7** Scheme illustrating the mechanism of impaired IL-4-induced M2a-subtype MΦ activation in obesity. IL-4/Irs2/Akt pathway was essential for activation of M2a-subtype MΦs, in addition to IL-4/STAT6 pathway. Both pathways were necessary for the full activation of M2a-subtype MΦs (left panel). In obesity, although IL-4/STAT6 pathway was maintained, IL-4/Irs2/Akt pathway was selectively impaired along with decreased *Irs2* expression and stabilization of the FoxO1/HDAC3/NCoR1 corepressor complex in MΦs (right panel)

in these mice, because PDK1 is a downstream molecule not only in the insulin signaling pathway, but also in the IL-4 signaling pathway. In fact, mice overexpressing CA-FoxO1 in the myeloid cells, which is thought to be associated with maintained *Irs2* expression and decreased IL-4 signaling due to stabilization of FoxO1/HDAC3/NCoR1 corepressor complex in the MΦs, showed reduced M2a-subtype MΦ activation and increased inflammation and insulin resistance under the HF diet condition[35]. These data suggest that IR signaling contributes to IL-4 signaling via regulating *Irs2* expression, mediating obesity-induced inflammation and insulin resistance under the HF diet condition.

Similar to the finding in the WAT, inflammation and insulin resistance were also noted in the livers of the M*Irs2*KO mice. Increased inflammatory cytokine release from the adipose tissue may cause hepatic insulin resistance. In fact, hepatic insulin resistance was observed in mice showing overexpression of *MCP-1* in the adipose tissue, along with increased release of inflammatory cytokines such as *TNFα* and *IL-6*[33]. +Kupffer cells, which are the resident MΦs in the liver, have been reported to contribute to hepatic inflammation and insulin resistance[5,32]. *PPAR-δ* deficiency has been shown to impair M2a-subtype MΦ activation in Kupffer cells, thereby aggravating hepatic inflammation and insulin resistance in obesity[11,12]. In addition to Kupffer cells, M1-type MΦs are recruited to the liver from the BM in obesity. The recruited hepatic MΦs are similar to the M1-type MΦs in the adipose tissue[36], and induce activation of proinflammatory pathways in the hepatocytes, thereby causing hepatic insulin resistance. Depletion of Kupffer cells and of the recruited hepatic MΦs using gadolinium or clodronate protected the mice from obesity-induced hepatic insulin resistance[37,38]. These data suggest three possibilities in respect of the mechanism underlying the hepatic inflammation and insulin resistance observed in M*Irs2*KO mice: increased inflammatory cytokine release from the adipose tissue, decreased M2a-subtype MΦ activation caused by *Irs2* deficiency in the Kupffer cells, and/or newly recruited M1-type MΦs in the liver.

Why was M2a-subtype MΦ activation maintained under the NC diet condition in spite of the fact this activation decreased under the HF diet condition in the M*Irs2*KO mice (Fig. 2a and Supplementary Fig. 2g)? There are two possibilities: induction of compensatory pathways in an IL-13-dependent or IL-13-independent manner. In type I receptor deficiency, M2a-subtype MΦ activation was indeed shown to be induced by IL-

13[39]. In type II receptor deficiency, on the other hand, IL-4 induced M2a-subtype MΦ activation[21], suggesting that both IL-4 and IL-13 activates M2a-subtype MΦs vice versa. Mice with systemic *STAT6* KO and myeloid-specific *IL-4Rα* KO, which show defect in both IL-4 and IL-13 signaling, exhibited a significant reduction of M2a-subtype MΦ activation in the ATMs under the NC diet condition[16]. In this study, although IL-4-induced M2a-subtype activation was impaired in the BMDMs of the M*Irs2*KO mice, IL-13-induced M2a-subtype activation was maintained in these cells. Jmjd3-IRF4 pathways have also been reported to induce M2a-subtype MΦ activation independently of the IL-13 signaling pathways[40]. These data suggest that the M2a-subtype MΦ activation observed in NC-fed M*Irs2*KO mice may be compensated for in both an IL-13-dependent and IL-13-independent manner.

In this study, we found that in obesity, the hyperinsulinemia presumably downregulates MΦ Irs2 expression, resulting in impaired IL-4-induced M2a-subtype MΦ activation, and consequently, development of inflammation and insulin resistance in the liver and WAT.

## Methods

**Animals**. To generate mice with targeted deletion of Irs2 in the myeloid lineage cells, mice with flanking loxP *Irs2* alleles (*Irs2*lox/lox) were crossed with LysozymeM-Cre transgenic mice[41,42]. Genotyping was performed by PCR amplification of the tail DNA from each mouse at 4 weeks of age, as previously reported[30]. To generate mice with targeted deletion of the *IR* in the myeloid lineage cells, we created *IR*lox/lox mice, carrying the IR allele with loxP sites flanking exon 4 (Supplementary Fig. 9a). The targeting construct to introduce loxP sites into the *IR* gene was created from a 13-kb clone of the IR gene containing exon 4, as described previously, but with slight modification[34]. A targeting vector with the floxed neomycin-resistance gene was introduced into the 5′ side of the IR gene, and the loxP gene was introduced into the 3′ side (Supplementary Fig. 9a). The construct was transfected into J1 embryonic stem cells (129/Sv) by electroporation and screened for homologous recombinant clones by Southern blot analysis. The cells were injected into blastocysts from C57BL/6 mice and transferred into pseudo-pregnant ICR female mice to generate heterozygous mice, as previously reported[41]. To generate M*IR*KO mice, *IR*lox/lox mice were crossed with LysozymeM-Cre transgenic mice. The PCR primers used for the Cre recombinase were 5′-ACATGTTCAGGGATCGCCAGG-3′ and 5′-TAACCAGTGAAACAGCATTGC-3′. To detect Cre-mediated recombination at the genomic DNA level in the BMDM by PCR, we designed primers for the upstream portion of the *IR* genomic DNA (primer a), Neo DNA (primer b), and the downstream portion of the *IR* genomic DNA (primer c). Primer a was 5′-TGCCTAGAGACTCCAAGACAAA-3′, primer b was 5′-CAGCGCATCGCCTTCTATCGCCTTC-3′ and primer c was 5′-CTGCAAAAAGGAGGAAATGC-3′, Primer pairs 'a' and 'b' or 'a' and 'c' yielded PCR products about 250 bp and 627 bp in length before and after Cre-mediated IR

deletion, respectively (Supplementary Fig. 9b). Although the original M$Irs2$KO and M$IR$KO mice were derived from a C57BL/6 and 129/Sv mixed background, these mice were backcrossed more than eight times with C57BL/6 mice. All experiments in this study were performed using male littermates, and $Irs2^{lox/lox}$ and $IR^{lox/lox}$ mice were used as the controls. The mice were housed under a 12-h light/dark cycle at 22.5–23.5 °C and given access to food *ad libitum*. Standard (CE-2) and HF (HF-32) diets were also purchased from Japan CLEA. The composition of the HF diet was: 20% safflower oil, 15.8% beef tallow, 24.5% casein, 6.75% sucrose, 5.5% cellulose, 5.0% mineral mixture, and 1.4% vitamin mixture. The HF diet was administered to the mice for 17–20 weeks starting at 8 weeks of age. The animal care and experimental procedures used in this study were approved by the Animal Care Committee of the University of Tokyo.

**Isolation of the peritoneal MΦs and flow cytometry**. Mice were injected intraperitoneally with 3% sodium thioglycollate (Wako). After four days, peritoneal MΦs were collected and stimulated with IL-4. Epididymal adipose tissue specimens isolated from mice were rinsed in PBS, minced into fine pieces, and digested with Tyrode buffer (130 mM NaCl, 5.4 mM KCl, 0.5 mM MgCl$_2$, 0.33 mM NaH$_2$PO$_4$, 22 mM glucose, 5 mM L-glutamine, and 25 mM HEPES) containing collagenase (Worthington) at 37 °C using a water bath for 15 min. Then, the samples were passed through a mesh and centrifuged in a swing rotor at $362 \times g$. The pellets were collected as the stromal-vascular fraction (SVF), and the SVF cells were incubated in 1× Pharm Lyse (BD Biosciences) for 8 min at room temperature. The cells were suspended in PBS containing 2% BSA, and incubated with anti-CD16/32 (553142, BD pharmingen) for 5 min on ice. Then, the cells were incubated with the primary antibodies or the matching control isotypes for 1 h on ice, and analyzed using a FACS Aria II cell sorter (BD Biosciences) and FlowJo (Tree Star, Ashland, OR). SiglecF-negative/F4/80-positive/CD11b-positive cells were sorted and used for the RNA extraction. Furthermore, the cells were also divided CD11c-positive/CD206-negative and CD11c-negative/CD206-positive cells, as M1-type and M2-type MΦs, respectively.

**Glucose and pyruvate tolerance test**. Mice were loaded with oral glucose at 1.5 mg/g body weight after being denied access to food for 24 h. Blood samples were taken at different time-points and the blood concentrations of glucose were measured with an automatic glucometer (Glutest Ace, Sanwa Chemical Co., Nagoya, Japan). Blood samples were collected and centrifuged in heparinized tubes, and the separated plasma samples were stored at −20 °C. Insulin levels were determined using a mouse insulin ELISA kit (Morinaga). Pyruvate tolerance test were performed after the animals were denied access to food for 16 h. The mice were injected intraperitoneally with pyruvate dissolved in saline (1.5 g/kg), and blood samples were obtained from the tail vein at different time-points.

**Hyperinsulinemic-euglycemic clamp**. An infusion catheter was inserted into the right jugular vein of the mice, as described previously[30]. To measure the GIR, a primed-continuous infusion of insulin (Humulin R; Lilly) was administered at the dose of 7.5 milliunits/kg/min to the HF-diet-fed mice, and the blood glucose concentration, monitored every 5 min, was maintained at approximately 120 mg/dl by the administration of glucose (5 g of glucose/10 ml enriched to about 20% with [6,6−$^2$H$_2$]glucose (Sigma)) for 120 min. Blood samples (20 µl) were obtained for 15 or 30 min before the end of the hyperinsulinemic-euglycemic clamp. Thereafter, the Rd was calculated according to nonsteady-state equations, and the EGP was calculated as the difference between the Rd value and the exogenous GIR.

**Insulin signaling in the liver, eWAT and skeletal muscle**. To investigate the insulin signaling in the liver, eWAT and skeletal muscle, insulin (6 ng/ml) (Humulin R; Lilly) was injected via the inferior vena cava. The liver, eWAT and skeletal muscle were dissected 10 min after the insulin infusion and immediately frozen in liquid nitrogen. The samples were then analyzed by western blot analysis.

**Hepatic TG content**. For determining the hepatic TG content, each liver sample was homogenized in buffer A (25 mM Tris-HCl, pH 7.4, 10 mM sodium ortho-vanadate, 10 mM sodium pyrophosphate, 100 mM sodium fluoride, 10 mM EDTA, 10 mM EGTA and 1 mM phenylmethylsulfonyl fluoride), as previously reported[31]. 2:1 (vol/vol) chloroform/methanol was added to the homogenate and the mixture was shaken for 15 min. After centrifugation at $17,753 \times g$ for 10 min, the organic layer was collected. This extraction was repeated three times, and the collected samples were dried and resuspended in 1% Triton X-100/ethanol. The TG content in the extracted sample was measured using Triglyceride E-test Wako (Wako Pure Chemical Industries Ltd., Osaka, Japan).

**Immunohistochemistry**. eWAT specimens were fixed with 10% formalin and embedded in paraffin. Quantification of the adipocyte size and F4/80-positive area were carried out using the image analyzer software, Win ROOF (Mitani Corp), as previously reported[41]. The BMDM were also fixed in 4% paraformaldehyde in PBS for 30 min on ice and blocked using 3% bovine serum albumin (BSA)/PBS for at least 1 h at room temperature. FoxO1 (2880, Cell signaling, 1:100) antibody was added, followed by incubation in blocking buffer (3% BSA/PBS) overnight at 4 °C.

The antibody reactivity was detected using Alexa Fluor 488-conjugated anti-rabbit IgG, a diaminobenzidine (DAB) substrate (Pierce). Fluorescence images were acquired using an Olympus IX71 microscope.

**Bone marrow-derived MΦs**. Bone marrow cells were isolated from the femur, tibia and humerus bones of the mice, as previously reported[30]. The cells, suspended in RPMI-1640 medium (supplemented with 15% FCS, 1% penicillin-streptomycin) were plated. The floating cells were collected after incubation for 16 h and plated at a concentration of $1 \times 10^6$ cells/ml in RPMI-1640 medium (supplemented with 15% FCS, 1% penicillin-streptomycin and 50 ng/ml recombinant M-CSF (Shenandoah biotechnology)) in 10-cm Petri dishes and allowed to differentiate for 8–10 days. For M1-type MΦ activation, the BMDM were treated with 5 ng/ml of LPS (Sigma), while for M2a-subtype MΦ activation, the cells were treated with 10 ng/ml of IL-4 (Peprotech) or 20 ng/ml of IL-13 (Peprotech). The cells were exposed to a PI3 kinase inhibitor (10 µM LY294002) 30 min before the addition of IL-4.

**Immunoprecipitation and western blot analysis**. To prepare the lysates, the cells and tissues were homogenized in buffer A (25 mM Tris-HCl, pH 7.4, 10 mM sodium orthovanadate, 10 mM sodium pyrophosphate, 100 mM sodium fluoride, 10 mM EDTA, 10 mM EGTA and 1 mM phenylmethylsulfonyl fluoride). For immunoprecipitation of Irβ, Irs1 and Irs2, the lysates were incubated with rabbit polyclonal antibody against Irβ (sc-711, Santa Cruz), Irs1 (06–248, Millipore) or Irs2 (3089, Cell Signaling) for 1 h at 4 °C. Then, protein G-Sepharose was added, followed by incubation for a further 1 h at 4 °C. Thereafter, after washing 3 times with buffer A, the immunocomplexes were resolved on 7% or 10% SDS-PAGE. Phosphorylated or total protein was analyzed by immunoblotting using specific antibodies against Irβ, Irs1, Irs2 and phosphotyrosine. Phosphorylated or total protein of Akt, FoxO1 and STAT6 was isolated by immunoblotting using specific antibodies after the tissue lysates were resolved by SDS-PAGE and transferred to a Hybond-P PVDF transfer membrane (Amersham Biosciences). Bound antibodies were detected with HRP-conjugated secondary antibodies using ECL detection reagents (Amersham Biosciences). Uncropped scans of the blots can be found as a Supplementary Figure in the Supplementary Information (Supplementary Fig. 10 and Supplementary Fig. 11).

**RNA extraction and real-time PCR**. RNA was isolated from the cells and tissues with the Qiagen RNeasy Kit (Qiagen, Germany), in accordance with the manufacturer's instructions. One microgram of RNA was used for generating cDNA using random hexamers with MultiScribe reverse transcription reagents (ABI). TaqMan quantitative PCR (cycles of 50 °C for 2 min, 95 °C for 10 min, followed by 40 cycles of 95 °C for 15 s, 60 °C for 1 min) was then performed with ABI Prism 7900 PCR (Applied Biosystems) to amplify *IR* (Mm 00439693_m1), *Irs1* (Mm00439720_s1), *Irs2* (Mm03038438_m1), *IL-4R* (Mm00439634_m1), *STAT6* (Mm01160477_m1), *Arg1* (Mm00475988_m1), *FIZZ1* (Mm00445109_m1), *Ym1* (Mm00657889_mH), *Mgl1* (Mm0054612_m1), *IL-10* (Mm00439614_m1), *TNFα* (Mm 00443258_m1), *MCP-1*(Mm00441242_m1), *IL-6* (Mm00446190_m1), *CCR2* (Mm99999051_gH), *IL-1β* (Mm00434228_m1), *IL-18* (Mm00434225_m1), *PEPCK* (Mm00440636_m1), *G6Pase* (Mm 00839363_m1), *ACC* (Mm01304287_m1), *FAS* (Mm01253300_g1), *SCD1* (Mm00772290_m1), *PPARγ* (Mm00440945_m1), *FSP27* (Mm00617672_m1), *CD36* (Mm 00432403_m1), *PGC1β* (Mm01258518_m1), *PPARδ* (Mm01305434_m1), *IL-12a* (Mm00434169_m1), *IL-12b* (Mm01288989_m1), and *β-actin* (Mm00607939_s1) cDNA from the samples. The primers for the other reactions were purchased from Applied Biosystems. The expression level of each of the transcripts was normalized to the constitutive expression level of *β-actin* mRNA.

**Arginase assays**. After the BMDM ($5.0 \times 10^6$ cells) were stimulated with IL-4 for 48 h, the cells were collected in ice-cold PBS. The cells were then lysed in buffer containing 0.4% Triton-X-100 and protease inhibitors, and the arginase activity were measured using the Arginase Assay Kit (Abnova), according to the manufacturer's instructions. The lysed cells were centrifuged at $14,000 \times g$ at 4 °C for 10 min, and the supernatants were plated on to a 96-well microtiter plate. L-arginine was converted to urea by a buffer containing a substrate and cofactor, and the absorbance of the samples was measured using a microplate reader at the wavelength of 430 nm.

**siRNA transfections and adenovirus-mediated gene transfer**. The BMDM were transfected with adenovirus containing murine cDNA encoding FoxO1-T24A/S253D/S316A[43]. Five days later, the cells were treated with IL-4 and the harvested samples were then used for the gene expression analyses. The Neon® Transfection System (Invitrogen) was used for the siRNA transfection. The BMDM were suspended at the density of $1.2 \times 10^7$ cells/ml in resuspension buffer T (Invitrogen), and incubated with 200 nM of siRNA (siFoxO1, 7892394, Invitrogen; siHDAC3, s67421, Ambion; siNCoR1, s73229, Ambion; siSTAT6, sc-36570, Sant Cruz; siCont, 4390843, Ambion). The pulse conditions were as follows: square wave, 1600 V, 1 pulse, 20 ms pulse length. The RNAi efficiency was confirmed by quantitative real-time PCR and the silenced BMDM were used for the functional assays 72–96 h after the electroporation. BMDM transfected with siRNA were stimulated with 10 ng/ml IL-4 for 20 h after pretreatment with 100 nM of insulin for 8 h.

**Dual luciferase assays**. Several DNA fragments containing the mouse *Arg1*, *FIZZ1* and *Ym1* promoters were PCR-amplified from mouse genomic DNA. The PCR products were ligated using the pGEM-T Easy vector (Promega) and their nucleotide sequences were verified by DNA sequencing. Mutation of the *Arg1* promoter was produced using the PrimerSTAR mutagenesis basal kit (Takara). These promoter fragments were cloned into the luciferase reporter pGL3-Basic vector (Promega, Madison, WI). Transfection was carried out at 70–80% confluence of the RAW264.7 cells (Cat. No. EC91062702, DS Pharma Biomedical) using 2.25 μg of the *Arg1*-, *FIZZ1*-, or *Ym1*-luciferase reporter gene (pGL3); 0.5 μg of the Renilla luciferase reporter vector (phRL-SV40; Promega) was used as the internal control for determining the transfection efficiency. After the cells were transfected using the Neon transfection system (Invitrogen), the transfected cells were treated with 10 ng/ml of IL-4 for 24 h. The luciferase activities were measured using the Dual-Glo Luciferase Assay System (Promega), according to the manufacturer's protocol.

**EMSA**. Nuclear protein extracts were prepared from RAW 267.4 cells before IL-4 treatment. A double-stranded oligonucleotide containing the consensus forkhead binding region in the *Arg1* promoter area was labeled using the Biotin 3′ End DNA Labeling Kit (PIERCE). This biotin-labeled oligonucleotide probe (20 fmol) was incubated with the nuclear extracts in the presence of 10 mM Tris-HCl (pH 7.5), 50 mM NaCl, 1 mM MgCl2, 0.5 mM EDTA, 4% glycerol and 0.5 mM DTT for 20 min at room temperature using the LightShift® Chemiluminescent EMSA Kit (Thermo scientific). For the competition experiments, the nuclear extracts were incubated with 0, 1, 2 or 5 μl of anti-FoxO1 (2880, Cell Signaling), anti-HDAC3 ((3949, Cell Signaling) or anti-NCoR1 (5948, Cell Signaling) antibody, or an equal amount of normal rabbit or IgG (sc-2027, Santa Cruz) for 60 min at room temperature. The non-denaturing binding reaction mixture was applied for electrophoresis on 6% polyacrylamide gels. The nylon membrane transferred from the gels was crosslinked using a commercial UV-light crosslinking instrument equipped with 254 nm bulbs and visualized by chemiluminescence.

**Co-culture system**. 3T3-L1 preadipocytes (ECACC 86052701) were purchased from DS Pharma Biomedical Co., Ltd. Co-culture was performed using the Transwell system (Merck Millipore) with a 0.4 μm porous membrane to separate the upper chambers from the lower ones, as previously described, with some modifications[12]. In the lower chambers, 3T3-L1 preadipocytes were seeded into 12-well plates at 8 × 10^4 cells per well and cultured to confluence in 3T3-L1 pre-adipocyte medium (DS Pharma Biomedical Co., Ltd). At 2 days post confluence, the cells were induced to differentiate using the 3T3-L1 differentiation medium (DS Pharma Biomedical Co., Ltd), and then used after 8 days. The BMDM of the control, M*Irs2*KO or C57BL/6 mice were plated at a concentration of 1 × 10^6 cells/ml in 10 cm Petri dishes containing RPMI-1640 medium and allowed to differentiate for 8 days. The BMDM (0.8 × 10^6 cells per well) were plated into the upper chambers. Twenty-four hours after the plating, FCS and M-CFS-free RPMI 1640 medium plus 100 nM insulin were added to the BMDM and 3T3-L1 adipocyte cultures. After further incubation for 24 h, the conditioned medium, 3T3-L1 cells and BMDM were collected and analyzed. The measurements of IL-4 (Abcam) and IL-13 (Cusabio) were conducted using mouse ELISA kits.

**ChIP assay**. BMDM (0.8 × 10^6 cells) were plated in 10-cm dishes and stimulated or not stimulated with 10 ng/ml of IL-4. The cells were fixed with 37% paraformaldehyde for 10 min at room temperature, and quenched with glycine for 5 min at room temperature. The collected cells were lysed using a buffer containing protease inhibitor from a shearing ChIP kit (Diagenode). The cells were sonicated for 18 cycles (30 s "ON", 30 s "OFF") at 200 W on ice using an ultrasonic homogenizer (Bioruptor UCW-310, Cosmo Bio Co.) to shear chromatin. Samples were diluted to 1.8 ml with the ChIP Dilution Buffer (50 mM Tris-HCL pH 8, 167 mM NaCl, 1.1% Triton X-100, 0.11% sodium deoxycholate, protease inhibitor cocktail) and precleared for 2 h at 4 °C with 50% protein G sepharose/salmon sperm DNA. After removal of the sepharose beads by centrifugation, immunoprecipitation was performed with ChIP-graded anti-FoxO1 (sc-11350, Santa Cruz, 1:50) HDAC3 (sc-11417, Santa Cruz, 1:50) and NCoR1 (sc-1609, Santa Cruz, 1:50) antibodies or an equal amount of normal rabbit (sc-2027, Santa Cruz, 1:50) or goat IgG (sc-2028, Santa Cruz, 1:50), followed by incubation overnight at 4 °C; then, precipitation of the antibody-protein-DNA complexes with 50% protein G sepharose/salmon sperm DNA was performed for 2 h at 4 °C. The precipitates were sequentially washed with buffers containing 0.1% SDS, 1% Triton X-100, 2 mM EDTA, and 20 mM Tris-HCl (pH 8.1) supplemented with either 150 mM (buffer I) or 500 mM NaCl (buffer II), prior to a final wash in 250 mM LiCl, 1% NP-40, 1% deoxycholate, 1 mM EDTA and 10 mM Tris-HCl (pH 8.1). The pellets were washed with Tris-EDTA buffer and extracted with 1% SDS, 10 mM EDTA and 50 mM Tris-HCl (pH 8.0). After heating at 65 °C overnight, the proteins were digested with proteinase K, and the DNA was purified. The samples were subjected to PCR using the following primers: for *Arg1*: 5′-GAATAGCACTTGGCACACGA-3′ and 5′-ACACTGTC-TAGGAAAGCATG-3′.

**Co-immunoprecipitation assay**. Stimulated BMDM were washed in ice-cold PBS, and harvested by scraping. The cells were centrifuged at 815 × *g* for 4 min at 4 °C, followed by addition of a co-immunoprecipitation (CO-IP) buffer (50 mM HEPES,

pH 8.0, 50 mM NaF, 10 mM Na4P2O7, 50 mM NaCl, 5 mM EDTA, 1 mM Na3VO4, 0.25% sodium deoxycholate, 1% NP-40 with a protease inhibitor cocktail (Roche)). The cells were suspended and placed on ice for 30 min. The samples were then centrifuged at 20,379 × *g* for 5 min at 4 °C, and the supernatants were saved as the cell lysates. The prepared cell lysates were incubated with each antibody for 3 h at 4 °C, followed by addition of protein G-sepharose beads (GE healthcare). After additional incubation for 1 h at 4 °C, the beads were washed three times with CO-IP buffer containing a protease inhibitor cocktail and resuspended in Laemmli buffer. The coimmunoprecipitated proteins were resolved by SDS-polyacrylamide gel electrophoresis (PAGE), and identified by western blot analysis. Uncropped scans of the blots can be found as a Supplementary Figure in the Supplementary Information (Supplementary Fig. 10).

**Antibodies**. For the western blot analysis, rabbit polyclonal antibody directed against Irβ (sc-711, Santa Cruz, 1:2000) and β-actin (A5441, SIGMA, 1:5000) was purchased from Santa Cruz Biotechnology. Rabbit polyclonal antibodies directed against Irs1 (06–248, Millipore, 1:2000) and Irs2 (MABS15, Millipore, 1:2000) and a mouse monoclonal antibody directed against phosphotyrosine (05–321, Millipore, 1:5000) were purchased from Millipore. Rabbit polyclonal antibodies directed against Akt (9272, Cell Signaling, 1:5000), phospho-Akt (Ser-473) (9271, Cell Signaling, 1:2000), phospho-FoxO1 (Ser-256) (9461, Cell Signaling, 1:1000), STAT6 (9362, Cell Signaling, 1:1000), FoxO1 (2880, Cell Signaling, 1:1000), Lamin B1 (9087, Cell Signaling, 1:2000) and G3PDH (2118, Cell Signaling, 1:2000) were purchased from Cell Signaling Technology. The rabbit polyclonal antibody directed against phospho-STAT6 (Tyr641) (06–937, Millipore, 1:1000) was purchased from Millipore. The antibodies used for the flow cytometry were SiglecF-PE (552126, BD Biosciences), CD11b-APC-Cy7 (557657, BD Biosciences), F4/80-FITC (11–4801, eBioscience), CD11c-eFluor405 (48–0114, eBioscience), and CD206-APC (141708, Biolegend). For the co-immunoprecipitation and EMSA, antibodies against HDAC3 (3949, Cell Signaling, 1:1000) and NCoR1 (5948, Cell Signaling, 1:1000) were used.

**Transwell migration assay**. Chemotaxis of BMDM was quantified using the Cultrex 24 well cell migration assay kit (Trevigen). The BMDM (0.5 × 10^6 /ml) were placed in the top chamber and 100 ng/ml of recombinant MCP-1 (R & D systems) was added at 0.5% FBS/RPMI to the bottom chamber. After incubation for 4 h at 37 °C, the cells in the bottom chamber were collected by cell dissociation. The collected cells were read at 520 nm emission and 485 nm excitation.

**Proliferation assay**. BMDM (0.2 × 10^5) were seeded in 96-well plates in volumes of 100 μl/well. After the cells were stimulated with 10 ng/ml of IL-4 for 24 h, BrdU was added at a final concentration of 1 μM. DNA synthesis was assayed by the Cell Proliferation ELISA, BrdU (Roche Molecular Biochemicals), using a luminometer (Promega).

**STZ plus phlorizin**. STZ (Sigma) was freshly suspended in sodium citrate buffer (pH 4.5). The suspension was injected intraperitoneally (150 mg/kg) on day 0 and day 5 to 8-week-old C57BL/6 mice that had been denied access to food for 5 h. Phlorizin (Wako) was dissolved in a solution containing 10% ethanol, 15% DMSO and 75% saline, and injected subcutaneously (0.4 g/kg) twice daily for 7 days starting from day 7 after the STZ injection. Peritoneal MΦs were then collected after injection of thioglycolate solution (Wako).

**Blood sample assay**. Plasma levels of TG, FFA, T-ch and HDL (Wako Pure Chemical Industries, Ltd) were assayed by enzymatic methods.

**Statistical Analysis**. Values are expressed as the mean ± SEM and were analyzed using the JMP 11 software (SAS Institute). Student's *t* test was used to analyze the statistical significances of differences between two groups, and ANOVA to analyze the statistical significances of differences among multiple groups. The Tukey-Kramer test was used for post hoc analysis. The statistical significance level was set at $P \le 0.05$ in all the tests.

## Data availability
The authors declare that all data supporting the findings of this study are available within the manuscript and its Supplementary Information files or are available from the author upon reasonable request.

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

## Acknowledgements

We thank Ayami Gouda, Yurika Shiraishi, Masatsugu Takayasu, Manami Takagi, Tamao Iwakami, Shiho Nemoto, Namiko Kasuga, Tomoko Asano, Eriko Nozaki and Kousuke Yokota for their excellent technical assistance and assistance with the animal care. This work was supported by a grant for TSBMI from the Ministry of Education, Culture, Sports, Science and Technology of Japan, a Grant-in-Aid for Scientific Research (A) (16209030), (A) (18209033), and (S) (20229008) from the Ministry of Education, Culture, Sports, Science, and Technology of Japan (to T. Kadowaki), a Grant-in-Aid for Scientific Research (C) (19591037) and (B) (21390279) from the Ministry of Education, Culture, Sports, Science, and Technology of Japan (to N.K.).

## Author contributions

T.Kubota, N.K and T.Kadowaki. designed this study and wrote the manuscript. T. Kubota, M.I., I.T., T.M., K.I. and K.T. M.M. conducted the experimental research and analyzed the data. M.M.,T.Y. and K.U. contributed to the data discussion. T.Kadowaki is the guarantor for this work, and as such, had full access to all the data in the study and takes responsibility for the integrity of the data and accuracy of the data analysis. All the authors gave their final approval for the manuscript version submitted for publication.

## Additional information

**Competing interests:** The authors declare no competing interests.

