## [Peer Review File · Nature Communications]

Reviewers' comments:

Reviewer #1 (Remarks to the Author):

In this manuscript entitled 'Inhibition of FoxO1/HDAC3/1 NCoR by IL-4/Irs2 is impaired in macrophages, causing decreased M2a-subtype activation and systemic insulin resistance in obesity', Kubota et al. describe the role of the IL4/IRS2 pathway in the development of obesity-induced inflammation and insulin resistance. By using various approaches including myeloid-specific deletion of IRS2 and IR, the authors conclude that the inhibition of the IL-4/IRS2 axis in macrophages leads to a reduced number of M2a macrophages translating into adipose tissue inflammation and insulin resistance in obesity. It is proposed that hyperinsulinemia promotes downregulation of IRS2 in adipose tissue macrophages leading to a reduced ability of adipose tissue macrophages to differentiate towards the anti-inflammatory M2a phenotype. The manuscript reads well and contains an impressive set of experiments. The data reveal a unique future of the IRS2 molecule in macrophages controlling their polarization potential in adipose tissue. However, the IL-4/IRS2 axis has already been identified as an important regulator of M2 differentiation. The novel part in this manuscript relates to the identification of hyperinsulinemia as suppressor of IRS2 mediated by the IR in macrophages. This in turn leads to a reduction in M2a polarized macrophages due to lower efficacy of IL-4.

Comments and suggestions

-As stated above, earlier papers have already established a role for the IL-4/IRS2 axis in the regulation of macrophage polarization. Several papers show similar findings related to the polarization of M2-like macrophages as reported in the current manuscript. What do the current results add, in a mechanistic way, to the existing data on the role of the IL-4/IRS2 axis in the regulation of M2a ?

-The m1 vs. m2 classification in the adipose tissue is a somewhat outdated concept. It would be very interesting if the authors could also demonstrate the importance of this pathway in metabolically activated macrophages that are known to reside in obese adipose tissue. Moreover, insulin has been shown to be one of the factors, along with glucose and palmitic acid, driving the metabolically activated macrophages in obese adipose tissue. This would be a very relevant issue for the authors to address.

-IRS2 ko in myeloid cells results in liver and adipose tissue specific insulin resistance without any differences on total BW. In adipose tissue of the IRS2 ko animals, more CLS were observed. The authors conclude, based upon data presented in figure 2, that the ratio of M1/M2 was increased in the ko animals and demonstrate that some markers of M2 macrophages were lower expressed in the ko animals. I am not really sure you can interpret these results as a reduction in M2a macrophages, since all data is reported as arbitrary units or percentage. The gene expression values and M1/M2 ration may actually reflect a strong upregulation in M1-like macrophages during HFD-feeding that may be accompanied by a less pronounced increase of M2 cells in the KO animals. In other words, I am not really convinced that the data presented in figure 2 would allow the authors to state that there is a significant reduction in M2a-like macrophages. It might also be an overwhelming influx of monocytes differentiating towards M1-like cells.

-Did the authors consider to measure IL-4 levels and eosinophils in the adipose tissue of the IRS2ko versus WT after HFD-feeding? Also, MCP-1 levels are higher in the KO animals suggestive of more influx into the adipose tissue that would account for more M1 cells not necessarily being accompanied by a reduction in M2a like cells.

-The authors go to great lengths to establish the contribution of the FOXO-NCoR1/HDAC3 corepressor complex being responsible for the suppression of the M2a-subtype (data presented in figure 3). However, do the authors have any data to support the in vivo relevance of this pathway? For example, what are the gene expression levels of these regulators in ATMs ? How are these factors expressed/regulated in adipose tissue macrophages? In addition, in an in vitro setting, does insulin treatment affect this pathway? Presumably, prolonged insulin exposure should directly impact on this pathway.

-I think the in vitro experiments using insulin together with IL-4 as presented in figure 5 are interesting. What is the concentration of insulin that was used in these experiments and how long

are the cells treated? On a more functional level, is the cytokine production of these cells also affected? Hence, do the cells produce different/lower levels of typical M2-like cytokines?

-I am a bit confused about certain figure legends used by the authors. For example, in figure 2g the authors have presented BMDMs from C57/bl6 animals, yet in the figure legends it reads 'in the C57/Bl6 mice after stimulation with IL-4'. This needs to be clarified.

-As explained by the authors in figure 7, hyperinsulinemia is the driving force in the development of inflammation and adipose tissue due to the dysregulation of M2a polarization. However, it has also been established that hyperinsulinemia drives expression of MCP-1 in adipose tissue leading to the recruitment of monocytes during obesity. Would it be possible to somehow lower plasma insulin levels (STZ treatment or metformin treatment for example) to reveal the potential contribution of hyperinsulinemia to the regulation of IRS2 levels in macrophages in vivo?

-Overall, I feel that most of the conclusions are justified based upon the presented data. However, some of the results are over interpreted and figure 7 is an oversimplified model. Currently, no data in the paper clearly supports the role of hyperinsulinemia in lowering IRS2 in macrophages.

Although the absence of the IR provides some proof, this would not explain why normal insulin levels would not affect the pathway and hyperinsulinemia would lead to downregulation of IRS2.

Reviewer #2 (Remarks to the Author):

Adipose tissue macrophages play a pivotal role for adipose tissue inflammation and insulin resistance in obesity, and macrophage subtypes can impact insulin sensitivity. This manuscript shows that 1) *Irs2* expression is downregulated in obese adipose tissue macrophages, 2) Myeloid-specific *Irs2* KO mice exhibit reduced M2 macrophage numbers, increased adipose tissue inflammation, and increased insulin resistance upon HFD feeding, 3) *Foxo1/HDAC3/NCoR1* inhibits M2-related gene expression, 4) Hyperinsulinemia represses *Irs2* expression and enhances *Foxo1* activity, which leads to repression of M2 marker genes.

Although many studies suggest the reduced number of M2 macrophages contribute to enhanced pro-inflammatory responses in adipose tissue inflammation and insulin resistance, the mechanisms for M2 macrophage reduction has not been clearly elucidated. This study provides new insights for how M2 macrophages may be reduced in obese adipose tissue. However, there are a few shortcomings that need attention:

1. The authors propose that reduced *Irs2* expression impairs M2a macrophage differentiation. Although they show the *Irs2* expression level in peritoneal macrophages and adipose tissue macrophages (SiglecF-CD11b+F4/80+), the downregulation of *Irs2* expression itself might be secondary to reduced M2 macrophages. Perhaps the expression levels of *Irs2* in M1 and M2 macrophages in adipose tissues in NCD and HFD fed mice could be determined. If there is no difference in *Irs2* expression in M2 macrophages in NCD-fed mice, it would be also helpful to explain no difference in M2 macrophage in NCD-fed myeloid-specific *Irs2* KO mice.

2. The authors show the alteration of the M2 population in MIRS2KO and MIRKO mice upon HFD. As IL-4 and IL-13 are key inducers for M2 macrophages, the level of IL-4 and IL-13 in their adipose tissues should be measured. Also, it would be better to show the marker gene expression or cell number of eosinophils in adipose tissues of HFD-fed MIRS2KO mice and HFD-fed MIRKO mice.

3. Regarding figure 2f, please show IL-4-induced Akt phosphorylation in BMDM from control mice and MIRS2 KO mice. The authors should show the *Irs2*-mediated Akt phosphorylation.

4. In figure 2j, it would be better to show IL-10 and M1 macrophage-related gene expression (including TNF- α , IL-1 β , IL-12) to compare the effect of *Irs2* and *Foxo1* in M1/M2a/M2b and c) differentiation. Also, what happens to M1-related gene expression after HDAC/NCoR1 knockdown in Figure 3f.

5. figure 3 shows the regulation of M2-related genes by *Foxo1/HDAC3/NCoR1* complex. To explore this complex in more depth, the promoter activity of *Arg1*, *FIZZ1*, and *Ym1* in CA-*Foxo1* with siHDAC3 and siNCoR1 transfected macrophages could be measured.

6. When the authors show the quantified data of pIrs2 protein expression, please normalize the phosphor-Irs2 by total Irs2 not β -actin. (Figure 1A and Figure 5B)

7. In Figure 2a and 4g, please label clearly on Y axis. Is it % of SVCs or % of F4/80+/CD11c+ cells? The % of M1 and M2 is higher than total macrophages. Also, is the liver TG level in mg/g tissue or mg/protein in supplementary figure 2e?

Responses to Reviewer #1

We are extremely grateful to Reviewer #1 for the very careful review of our manuscript.

Comments and suggestions

Concerning the experimental setup, the authors performed the co-culturing using only 8000 adipocytes together with 8 million BMDMs as stated in the M&M? These cell numbers seem very odd, even more in light of the reported IL-4 in the conditioned medium produced by the adipocytes and not the macrophages. Can the authors comment on this?

Also, the authors use the co-culture system to study if insulin downregulates macrophage Irs2 expression levels and subsequently increases the levels of inflammatory cytokines in the adipose tissue. How was this experiment being performed? Was insulin added to the co-culture system and thus affecting both macrophages and adipocytes? This is of importance since the authors conclude that insulin downregulates macrophage Irs2 downregulation and increases the levels of inflammatory cytokines in the adipose tissue. I guess the authors refer to the results in adipocytes part of the co-culture system showing reduced inflammatory gene expression levels. Hence, the authors should provide more experimental details and be cautious when interpreting the result.

We thank you for paying careful attention to our revised manuscript. We apologize for our inaccurate and insufficient description of the co-culture system. First, as pointed out by the reviewer, we mistakenly stated that “differentiated 3T3-L1 cells (0.8×10^4 cells) were cultured in the lower chamber, while the BMDM (0.8×10^7 cells) of control, MIrs2KO or C57BL/6 mice were cultured in the upper chamber”. We should have stated that “in the lower chambers, 3T3-L1 preadipocytes were seeded into 12-well plates at 0.8×10^4 cells per well and cultured to confluence in 3T3-L1 preadipocyte medium (DS Pharma Biomedical Co., Ltd). The BMDM (0.8×10^6 cells per well) were plated into the upper chambers.” Second, as pointed out by the reviewer, insulin added to the co-culture system could affect both the macrophages and adipocytes. In accordance with the reviewer’s suggestion, we have given a detailed description of the co-culture system in Experimental Procedures, as follows (Page 28, lines 630-643 in the second revised version of our manuscript);

“3T3-L1 preadipocytes (ECACC 86052701) were purchased from DS Pharma Biomedical Co., Ltd. Co-culture was performed using the Transwell system (Merck Millipore) with a 0.4 μm porous membrane to separate the upper chambers from the lower ones, as previously described, with some modifications¹². In the lower chambers, 3T3-L1 preadipocytes were seeded into 12-well plates at 0.8×10^4 cells per well and cultured to confluence in 3T3-L1 preadipocyte medium (DS Pharma Biomedical Co., Ltd). At 2 days post confluence, the cells were induced to differentiate using the 3T3-L1 differentiation medium (DS Pharma Biomedical Co., Ltd), and then used after 8 days. The BMDM of the control, MIRS2KO or C57BL/6 mice were plated at a concentration of 1×10^6 cells/ml in 10-cm Petri dishes containing RPMI-1640 medium and allowed to differentiate for 8 days. The BMDM (0.8×10^6 cells per well) were plated into the upper chambers. Twenty-four hours after the plating, FCS and M-CFS-free RPMI 1640 medium plus 100 nM insulin were added to the BMDM and 3T3-L1 adipocyte cultures. After further incubation for 24 hours, the conditioned medium, 3T3-L1 cells and BMDM were collected and analyzed. The measurements of IL-4 (Abcam) and IL-13 (Cusabio) were conducted using mouse ELISA kits.”

The reviewer is correct in that we referred to the results in adipocytes part of the co-culture system. In the first revised version of our manuscript, we compared MCP-1, CCR2, IL-18 and IL-6 expression levels of 3T3-L1 cells co-cultured with BMDM in the presence of insulin, with those in the absence of insulin. In order to assess the effects of insulin-pretreated BMDM on 3T3-L1 cells, we should have included the data of insulin-pretreated 3T3-L1 cells without BMDM. In accordance with the suggestion of the reviewer, we have added data about the inflammatory cytokine levels in the insulin-pretreated 3T3-L1 cells without BMDM to Figure 5f and Supplementary Fig. 7f in the second revised version of our manuscript, as follows (Page 15, lines 330-333 in the second revised version of our manuscript).

“Moreover, when the BMDM were co-cultured with 3T3-L1 cells in the presence of insulin, significant downregulation of the Irs2 expression in the BMDM and significant upregulation of MCP-1, CCR2, IL-18 and IL-6 expressions in the 3T3-L1 cells were observed (Fig. 5f and Supplementary Fig. 7f, g).”

REVIEWERS' COMMENTS:

Reviewer #1 (Remarks to the Author):

The authors have responded adequately to my comments.